# Tackling Data Corruption in Offline Reinforcement Learning via Sequence Modeling

**Jiawei Xu**[1][*] **Rui Yang**[2][*]  **Shuang Qiu**[3]  **Feng Luo**[4]  **Meng Fang**[5]  **Baoxiang Wang**[1]  **Lei Han**[6]

[1]The Chinese University of Hong Kong, Shenzhen  [2]University of Illinois Urbana-Champaign
[3]City University of Hong Kong  [4]Rice University  [5]University of Liverpool  [6]Tencent Robotics X

## Abstract

Learning policy from offline datasets through offline reinforcement learning (RL) holds promise for scaling data-driven decision-making while avoiding unsafe and costly online interactions. However, real-world data collected from sensors or humans often contains noise and errors, posing a significant challenge for existing offline RL methods, particularly when the real-world data is limited. Our study reveals that prior research focusing on adapting predominant offline RL methods based on temporal difference learning still falls short under data corruption when the dataset is limited. In contrast, we discover that vanilla sequence modeling methods, such as Decision Transformer, exhibit robustness against data corruption, even without specialized modifications. To unlock the full potential of sequence modeling, we propose **R**obust **D**ecision **T**ransformer (**RDT**) by incorporating three simple yet effective robust techniques: embedding dropout to improve the model's robustness against erroneous inputs, Gaussian weighted learning to mitigate the effects of corrupted labels, and iterative data correction to eliminate corrupted data from the source. Extensive experiments on MuJoCo, Kitchen, and Adroit tasks demonstrate RDT's superior performance under various data corruption scenarios compared to prior methods. Furthermore, RDT exhibits remarkable robustness in a more challenging setting that combines training-time data corruption with test-time observation perturbations. These results highlight the potential of sequence modeling for learning from noisy or corrupted offline datasets, thereby promoting the reliable application of offline RL in real-world scenarios. Our code is available at https://github.com/jiawei415/RobustDecisionTransformer.

## 1 Introduction

Offline reinforcement learning (RL) aims to derive near-optimal policies from fully offline datasets (Levine et al., 2020; Kumar et al., 2020; Fujimoto et al., 2019; Wang et al., 2018), thereby reducing the need for costly and potentially unsafe online interactions with the environment. However, offline RL encounters a significant challenge known as distribution shift (Levine et al., 2020), which can lead to performance degradation. To address this challenge, several offline RL algorithms impose policy constraints (Wang et al., 2018; Fujimoto et al., 2019; Fujimoto & Gu, 2021; Kostrikov et al., 2021; Park et al., 2024) or maintain pessimistic values for out-of-distribution (OOD) actions (Kumar et al., 2020; An et al., 2021; Bai et al., 2022; Yang et al., 2022a; Ghasemipour et al., 2022), ensuring that the learned policy aligns closely with the training distribution. Although the majority of traditional offline RL methods rely on temporal difference learning, an alternative promising paradigm for offline RL emerges in the form of sequence modeling (Chen et al., 2021; Janner et al., 2021; Shi et al., 2023; Wu et al., 2024). Unlike traditional RL methods, Decision Transformer (DT) (Chen et al., 2021), a representative sequence modeling method, treats offline RL as a supervised learning task, predicting actions directly from sequences of reward-to-gos, states, and actions. Prior work (Bhargava et al., 2023) suggests that DT excels at handling tasks with sparse rewards and suboptimal quality data, showcasing the potential of sequence modeling for real-world applications.

---

[*]Equal Contribution. Correspondence to Jiawei Xu⟨jiawei202016@gmail.com⟩ and Rui Yang⟨yangrui.thu2015@gmail.com⟩.

When deploying offline RL in practical settings, dealing with noisy or corrupted data resulting from data collection or malicious attacks is inevitable (Zhang et al., 2020; 2021; Liang et al., 2024; Yang et al., 2024a). Consequently, robust policy learning from such data is essential for the successful deployment of offline RL. A series of prior works (Zhang et al., 2022; Ye et al., 2024b; Wu et al., 2022; Chen et al., 2024; Ye et al., 2024a) focus on the theoretical properties and certification of offline RL under data corruption. Notably, Ye et al. (2024b) propose an uncertainty-weighted offline RL algorithm with Q ensembles to address reward and dynamics corruption, while Yang et al. (2024c) enhance the robustness of Implicit Q-Learning (IQL) (Kostrikov et al., 2021) against corruptions on all elements using Huber loss and quantile Q estimators. These advancements have primarily concentrated on temporal difference learning, dedicating significant effort to learning robust Q functions from corrupted data. This approach brings up an intriguing question: by bypassing the Q function, *can sequence modeling methods effectively handle data corruption in offline RL?*

In this study, we conduct a comparative analysis between prior robust offline RL methods and DT (without any robust modifications) under various data corruption settings. Surprisingly, we find that traditional methods relying on temporal difference learning struggle in settings with limited data and significantly underperform compared to DT. Additionally, DT demonstrates superior performance over previous methods in scenarios involving state attacks. These findings highlight the promising potential of sequence modeling for addressing data corruption challenges in offline RL. To further unlock the capabilities of sequence modeling methods under data corruption, inspired by successes in the supervised learning domain (Song et al., 2022; Chang et al., 2017; Reed et al., 2014; Gal & Ghahramani, 2016), we propose **R**obust **D**ecision **T**ransformer (**RDT**). RDT aims to boost the robustness of DT in three aspects: model structure, loss function, and data refinement. To achieve this, RDT incorporates three simple yet highly effective techniques to mitigate the impact of corrupted data: *embedding dropout*, *Gaussian weighted learning*, and *iterative data correction*. These simple modifications are designed to address the fundamental limitations of the standard DT and unlock the potential of sequence modeling for data corruption.

To comprehensively study RDT's robustness under data corruption with limited data, we constructed a variety of small datasets from MuJoCo, Kitchen, and Adroit, creating a challenging benchmark for current robust offline RL methods. Through our experiments, we demonstrate that RDT outperforms conventional temporal difference learning and sequence modeling approaches under both random and adversarial data corruption scenarios, achieving a substantial 29% improvement in overall performance over DT. Furthermore, we show that after learning from corrupted data, RDT exhibits remarkable robustness against testing-time observation perturbations, indicating its potential to address both training-time and test-time attacks simultaneously. Our study emphasizes the significance of robust sequence modeling and offers valuable insights for the trustworthy deployment of offline RL in real-world applications.

## 2 PRELIMINARIES

**RL and Offline RL.** RL is generally formulated as a Markov Decision Process (MDP) defined by a tuple $(S, A, P, r, \gamma)$. This tuple comprises a state space $S$, an action space $A$, a transition function $P$, a reward function $r$, and a discount factor $\gamma \in [0, 1]$. The objective of RL is to learn a policy $\pi(a|s)$ that maximizes the expected cumulative return: $\max_\pi \mathbb{E}_{s_0 \sim \rho_0, a_t \sim \pi(\cdot|s_t), s_{t+1} \sim P(\cdot|s_t, a_t)} \left[ \sum_{t=0}^{T-1} \gamma^t r(s_t, a_t) \right]$, where $\rho_0$ denotes the distribution of initial states and $T$ is the trajectory length. In offline RL, the objective is to optimize the RL objective with a previously collected dataset $\mathcal{D} = \{(s_t^{(i)}, a_t^{(i)}, r_t^{(i)}, s_{t+1}^{(i)})_{t=0}^{T-1}\}_{i=0}^{N-1}$, which contains a total of $N$ trajectories. The agent cannot directly interact with the environment during the offline phase.

**Decision Transformer (DT).** DT models decision-making from offline datasets as a sequence modeling problem. The $i$-th trajectory $\tau^{(i)}$ of length $T$ in dataset $\mathcal{D}$ is reorganized into a sequence of return-to-go $R_t^{(i)}$, state $s_t^{(i)}$, action $a_t^{(i)}$:

$$\tau^{(i)} = \left( R_0^{(i)}, s_0^{(i)}, a_0^{(i)}, \ldots, R_{T-1}^{(i)}, s_{T-1}^{(i)}, a_{T-1}^{(i)} \right). \tag{1}$$

Here, the return-to-go $R_t^{(i)}$ is defined as the sum of rewards from the current step to the end of the trajectory: $R_t^{(i)} = \sum_{t'=t}^{T} r_{t'}^{(i)}$. DT employs three linear projection layers to project the return-

to-gos, states, and actions to the embedding dimension, with an additional learned embedding for each timestep added to each token. A GPT model is adopted by DT to autoregressively predict the actions $a_t^{(i)}$ with input sequences of length $K$:

$$\mathcal{L}_{DT}(\theta) = \mathbb{E}_{\tau^{(i)} \sim \mathcal{D}} \left[ \frac{1}{K} \sum_{t=0}^{K-1} \|\pi_\theta(\tau_{t-K+1:t-1}^{(i)}, R_t^{(i)}, s_t^{(i)}) - a_t^{(i)}\|_2^2 \right], \qquad (2)$$

where $\tau_{t-K+1:t-1}^{(i)}$ indicates the segment of $\tau^{(i)}$ from timestep $t - K + 1$ to $t - 1$.

**Data Corruption in Offline RL.** In prior works (Ye et al., 2024b; Yang et al., 2024c), the data is stored in transitions, and data corruption is performed on individual elements (state, action, reward, next-state) of each transition. This approach does not align well with trajectory-based sequence modeling methods like DT. In this paper, we consider a unified trajectory-based storage, where corrupting a next-state in a transition corresponds to corrupting a state in the subsequent transition, while corruption for rewards and actions is consistent with prior works. To elaborate, an original trajectory is denoted as $\tau_{\text{origin}}^{(i)} = \left( s_0^{(i)}, a_0^{(i)}, r_0^{(i)}, \ldots, s_{T-1}^{(i)}, a_{T-1}^{(i)}, r_{T-1}^{(i)} \right)$, which can be reorganized into the sequence data of DT in Eq. 1 or split into $T - 1$ transitions $\left( s_t^{(i)}, a_t^{(i)}, r_t^{(i)}, s_{t+1}^{(i)} \right)_{t=0}^{T-2}$ for MDP-based methods. Note that here are only three independent elements (i.e., states, actions, and rewards) under the trajectory-based storage formulation.

Data corruption injects random or adversarial noise into the original states, actions, and rewards. **Random corruption** adds random noise to the affected elements in the datasets, resulting in a corrupted trajectory denoted as $\tau_{\text{corrupt}}^{(i)} = \left( \hat{s}_0^{(i)}, \hat{a}_0^{(i)}, \hat{r}_0^{(i)}, \ldots, \hat{s}_{T-1}^{(i)}, \hat{a}_{T-1}^{(i)}, \hat{r}_{T-1}^{(i)} \right)$. For instance, $\hat{s}_0^{(i)} = s_0^{(i)} + \lambda \cdot \text{std}(s), \lambda \sim \text{Uniform}[-\epsilon, \epsilon]^{d_s}$ ($d_s$ is the dimensions of state, and $\epsilon$ is the corruption scale) and $\text{std}(s)$ is the $d_s$-dimensional standard deviation of all states in the offline dataset. In contrast, **adversarial corruption** uses Projected Gradient Descent attack (Madry et al., 2017) with pretrained value functions. Specifically, we introduce learnable noise to the states or actions and then optimize this noise by minimizing the pretrained value functions through gradient descent. More details about data corruption are provided in Appendix C.1.

## 3 SEQUENCE MODELING FOR OFFLINE RL WITH DATA CORRUPTION

We aim to answer the question of whether sequence modeling methods can effectively handle data corruption in offline RL in Section 3.1. To achieve this, we compare DT with prior offline RL methods in the context of data corruption. Based on the insights gained from the motivating example, we then propose enhancements for improving the robustness of DT in Section 3.2.

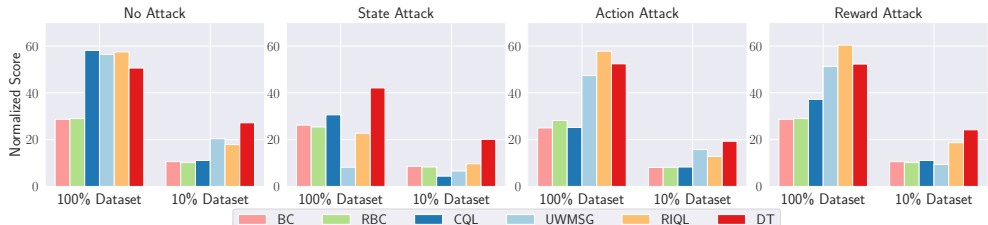

Figure 1: Average normalized scores of offline RL algorithms under random data corruption across three MuJoCo tasks (halfcheetah, walker2d, and hopper) using "medium-replay-v2" datasets. Many offline RL algorithms experience substantial performance declines when subjected to data corruption. In contrast, DT demonstrated remarkable robustness, particularly in the $10\%$ data regime.

### 3.1 MOTIVATING EXAMPLE

As illustrated in Figure 1, we compare DT with various offline RL algorithms under data corruption. We apply random corruption introduced in Section 2 on states, actions, and rewards. Specifically, we perturb $30\%$ of the transitions in the dataset and introduce random noise at a scale of 1.0 standard

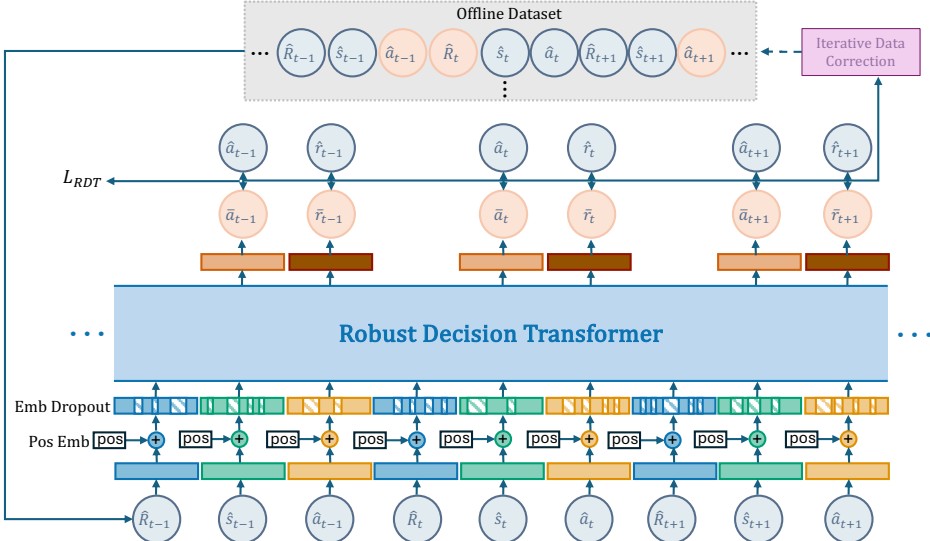

Figure 2: Framework of Robust Decision Transformer (RDT). RDT enhances the robustness of DT against data corruption by incorporating three components on top of DT: embedding dropout, Gaussian weighted learning, and iterative data correction.

deviation. In addition to the conventional full dataset setting employed in previous studies (Ye et al., 2024b; Yang et al., 2024c), we also explore a scenario with a reduced dataset, comprising only $10\%$ of the original trajectories. Our results reveal that most offline RL methods are highly vulnerable to data corruption, particularly in more challenging, limited data settings. In contrast, DT exhibits remarkable robustness to data corruption, especially in scenarios involving state corruption and limited dataset conditions. This resilience can be attributed to DT's formulation based on sequence modeling and its supervised training paradigm. We will delve deeper into the critical components affecting DT's robustness in Appendix D.1.

Overall, the results demonstrate DT can outperform prior imitation-based and temporal difference based methods without any robustness enhancement. This observation raises an interesting question:

***How can we further unleash the potential of sequence modeling in addressing data corruption with limited dataset in offline RL?***

## 3.2 ROBUST DECISION TRANSFORMER

To improve the robustness of DT against various data corruptions, we draw inspiration from successes in the supervised learning domain (Song et al., 2022; Chang et al., 2017; Reed et al., 2014; Gal & Ghahramani, 2016) to introduce the Robust Decision Transformer (RDT). **RDT enhances DT across three key aspects: model structure, loss function, and data refinement**. This is achieved through the incorporation of three simple yet effective components: embedding dropout (Section 3.2.1), Gaussian weighted learning (Section 3.2.2), and iterative data correction (Section 3.2.3). These simple modifications are designed to tackle the fundamental limitations of the standard DT and help unlock the potential of sequence modeling for data corruption. The overall framework is illustrated in Figure 2. Notably, RDT predicts both actions and rewards (instead of reward-to-gos). Given the typically high dimensionality of states, we avoid predicting states to mitigate potential negative impacts as discussed in Appendix D.14. Additionally, rewards provide more direct supervision for the policy compared to reward-to-gos, which also depend on future actions.

### 3.2.1 EMBEDDING DROPOUT

In the context of data corruption, corrupted states, actions, and rewards can introduce shifted inputs or erroneous features to the model. This can lead to a performance drop if the model overfits these harmful features. Therefore, developing a robust representation is crucial to enhance the model's resilience against data corruption.

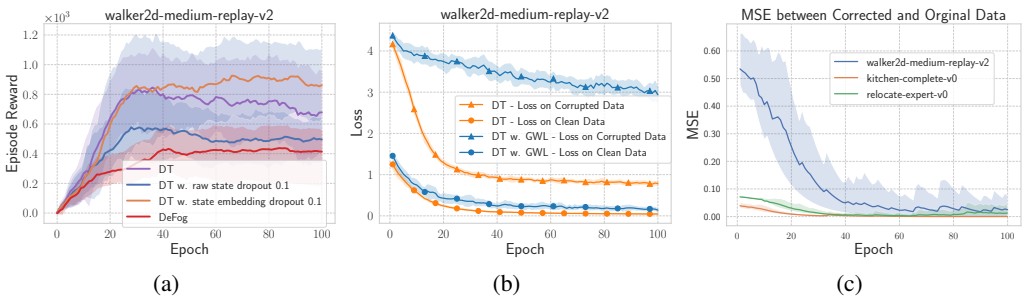

Figure 3: **(a) Comparing dropout methods under state attack:** Embedding dropout outperforms directly dropping the entire state (DeFog) or dropping dimensions on the raw state. **(b) Gaussian weighted learning under action attack:** Gaussian weighted learning (DT w. GWL) alleviates overfitting to the corrupted data and slightly minimizes the loss on clean data. **(c) Iterative data correction (DT w. IDC ) under action attack:** The MSE between corrected and oracle data gradually decreases to near zero.

Several studies (Carroll et al., 2022; Hu et al., 2023) utilize element-wise masking to learn robust representations by randomly masking state, return-to-go, or action elements in a trajectory, leveraging the redundancy in the trajectory's information. **However, we discovered that corrupted trajectories contain less redundant information. Discarding raw elements directly can lead to notable information loss, ultimately resulting in decreased performance (see Figure 3(a)).** To address this, we propose randomly dropping dimensions in the feature space rather than masking raw elements. Specifically, we employ embedding dropout, which encourages the model to learn more robust embedding representations while preventing overfitting (Merity et al., 2017).

To clarify, we define three hidden embeddings after the linear projection layer as $h_{R_t}$, $h_{s_t}$, and $h_{a_t}$, formulated as:

$$h_{R_t} = \phi_R(R_t) + \phi_t(t), \quad h_{s_t} = \phi_s(s_t) + \phi_t(t), \quad h_{a_t} = \phi_a(a_t) + \phi_t(t). \tag{3}$$

Here, $\phi_R$, $\phi_s$, and $\phi_a$ denote linear projection layers on different elements, while $\phi_t(t)$ represents the time-step embedding. Subsequently, we apply randomized dimension dropping on these feature embeddings, resulting in the corresponding masked feature embeddings as follows:

$$\tilde{h}_{R_t} = \mathcal{M}(p) \odot h_{R_t}, \quad \tilde{h}_{s_t} = \mathcal{M}(p) \odot h_{s_t}, \quad \tilde{h}_{a_t} = \mathcal{M}(p) \odot h_{a_t}. \tag{4}$$

The function $\mathcal{M}(\cdot)$ takes a probability $p \in [0, 1]$ as input and outputs a binary mask with the same shape as embeddings to determine the dimension inclusion. We apply the same drop probabilities $p$ within a moderate range $(0.1, 0.3)$ for different element embeddings.

### 3.2.2 GAUSSIAN WEIGHTED LEARNING

In addition to mitigating the negative impact of erroneous inputs, it is also essential to address the influence of corrupted labels. In DT, actions serve as the most crucial supervised signals, acting as the labels. Therefore, erroneous actions can directly influence the model through backpropagation. In RDT, we predict both actions and rewards, with rewards helping to reduce overfitting to corrupted action labels. To further diminish the impact of corrupt data, we focus on minimizing the influence of unconfident action and reward labels that could misguide policy learning.

To identify uncertain labels, we adopt a simple but effective method: we use the value of the sample-wise loss to adjust the weight for the DT loss. **The underlying insight is that a corrupted label would typically lead to a larger loss (see Figure 3(b)).** To softly reduce the effect of potentially corrupted labels, we use the Gaussian weight, i.e., a weight that decays exponentially in accordance with the sample's loss. This is mathematically formulated as:

$$w_{a_t}^{(i)} = e^{-\beta_a \cdot \delta_{a_t}^2}, w_{r_t}^{(i)} = e^{-\beta_r \cdot \delta_{r_t}^2}, \text{ where } \delta_{a_t} = \text{no\_grad}(\|\pi_\theta(\hat{\tau}_{t-K+1:t-1}^{(i)}, \hat{R}_t^{(i)}, \hat{s}_t^{(i)}) - \hat{a}_t^{(i)}\|_2),$$

$$\delta_{r_t} = \text{no\_grad}(\|\pi_\theta(\hat{\tau}_{t-K+1:t-1}^{(i)}, \hat{R}_t^{(i)}, \hat{s}_t^{(i)}, \hat{a}_t^{(i)}) - \hat{r}_t^{(i)}\|_2).$$

$$\tag{5}$$

In Eq.5, $\delta_{a_t}$ and $\delta_{r_t}$ represent prediction errors at step $t$ with detached gradients. The variables $\beta_a \geq 0, \beta_r \geq 0$ act as the temperature coefficients, providing flexibility to control the "blurring" effect of the Gaussian weights. Different tasks exhibit preferences for different temperature coefficients, as

demonstrated in Appendix D.3. With a larger value, more samples would be down-weighted. The loss function of RDT is expressed as follows:

$$
\mathcal{L}_{RDT}(\theta) = \mathbb{E}_{\hat{\tau}^{(i)} \sim \mathcal{D}} \left[ \frac{1}{K} \sum_{t=0}^{K-1} \left[ w_{a_t}^{(i)} \| \pi_\theta(\hat{\tau}_{t-K+1:t-1}^{(i)}, \hat{R}_t^{(i)}, \hat{s}_t^{(i)}) - \hat{a}_t^{(i)} \|_2^2 \right. \right.
$$

$$
\left. \left. + w_{r_t}^{(i)} \| \pi_\theta(\hat{\tau}_{t-K+1:t-1}^{(i)}, \hat{R}_t^{(i)}, \hat{s}_t^{(i)}, \hat{a}_t^{(i)}) - \hat{r}_t^{(i)} \|_2^2 \right] \right]. \tag{6}
$$

Gaussian weighted learning enables us to mitigate the detrimental effects of corrupted labels, thereby enhancing the algorithm's robustness.

### 3.2.3 ITERATIVE DATA CORRECTION

While the first two techniques have significantly reduced the impact of corrupted data, these erroneous data can still affect the policy as they remain in the datasets. We propose iteratively correcting corrupted data in the dataset using the model's predictions to bring the data closer to their true values in the next iteration. This method can further minimize the detrimental effects of corruption and can be implemented to correct reward-to-go, state, and action elements. In RDT, it is straightforward to correct the actions and rewards in the dataset and recalculate the reward-to-gos using corrected rewards. Therefore, our implementation focuses on correcting actions and reward-to-gos in the datasets, leaving better data correction methods for states for future work.

Initially, we store the distribution information of prediction error $\delta$ in Eq.5 throughout the learning phase to preserve the mean $\mu_\delta$ and variance $\sigma_\delta^2$ of actions and rewards, updating them with every batch of samples. The hypothesis is that the prediction error $\delta$ between predicted and clean labels should exhibit consistency after sufficient training. Therefore, if erroneous label actions are encountered, $\delta$ will deviate from the mean $\mu_\delta$, behaving like outliers. This deviation essentially enables us to detect and correct the corrupted data using the information of $\delta$.

Taking actions as an example, to detect corrupted actions, we calculate the $z$-score, denoted by $z^{(i)} = \frac{\delta^{(i)} - \mu_\delta}{\sigma_\delta}$, for each sampled action $\hat{a}^{(i)}$. If the condition $z^{(i)} > \zeta \cdot \sigma_\delta$ is met for any given action $\hat{a}^{(i)}$, we infer that the action $\hat{a}^{(i)}$ has been corrupted. We then permanently replace $\hat{a}^{(i)}$ in the dataset with the predicted action. This helps eliminate corrupted actions from the dataset (see Figure 3(c)). Correcting rewards follows a similar process, with the additional step of recalculating the reward-to-gos. The hyperparameter $\zeta$ determines the detection thresholds: a smaller $\zeta$ will classify more samples as corrupted data, while a larger $\zeta$ results in fewer modifications to the dataset. Empirically, we can achieve performance improvement by setting $\zeta$ to approximately 5. More implementation details of RDT can be found in Appendix C.3.

## 4 EXPERIMENTS

In this section, we conduct a comprehensive empirical evaluation of RDT, focusing on the following key questions: **1)** How does RDT perform under different data corruption scenarios? **2)** Is RDT robust to observation perturbations during the testing phase? **3)** What is the contribution of each component of RDT to its overall performance? **4)** How do baselines compare to RDT when equipped with a transformer backbone?

### 4.1 EXPERIMENTAL SETUPS

We evaluate RDT using various offline RL benchmarks, such as MuJoCo, KitChen, and Adroit (Fu et al., 2020). Two types of data corruption during the training phase are simulated: **random** and **adversarial corruption**, attacking states, actions, and rewards. Specifically, the corruption rate is set to 30%, and the corruption scale is $\epsilon = 1.0$ for the main results , similar to previous work (Yang et al., 2024c). Data corruption is introduced in Section 2, and more details can be found in Appendix C.1. We find that the data corruption problem can be exacerbated when the data is limited. To simulate such a scenario, we down-sample on both MuJoCo and Adroit tasks, referred to as MuJoCo (10%) and Adroit (1%), Specifically, we randomly select 10% (and 1%) of the trajectories from MuJoCo

Table 1: Detailed comparative results under random data corruption.

| Attack | Task | BC | RBC | DeFog | CQL | UWMSG | RIQL | DT | **RDT** |
|---|---|---|---|---|---|---|---|---|---|
| State | halfcheetah (10%) | 1.9±0.2 | 1.7±0.5 | 6.7±0.5 | **9.7±0.8** | 3.9±0.5 | 4.4±0.9 | 6.3±0.4 | 8.3±1.5 |
| | hopper (10%) | 16.4±3.7 | 13.0±2.3 | 23.1±5.5 | 2.2±1.5 | 16.3±7.4 | 15.5±5.4 | 36.1±7.6 | **40.8±3.5** |
| | walker2d (10%) | 7.5±0.9 | 10.3±1.4 | 9.0±3.3 | 1.2±1.8 | -0.4±0.2 | 9.2±4.4 | 18.0±2.5 | **20.3±2.8** |
| | kitchen-complete | 15.9±5.2 | 21.1±3.1 | 37.1±13.4 | 12.9±4.3 | 0.0±0.0 | 37.5±6.4 | 37.0±6.2 | **52.8±1.8** |
| | kitchen-partial | 17.1±2.6 | 20.2±2.8 | 9.8±6.6 | 0.0±0.0 | 0.0±0.0 | 25.9±3.4 | 31.0±8.1 | **36.8±5.8** |
| | kitchen-mixed | 17.4±4.9 | 29.2±5.6 | 10.6±5.0 | 0.0±0.0 | 0.0±0.0 | 21.6±3.7 | 31.8±3.4 | **41.8±4.3** |
| | door (1%) | 75.7±11.7 | 73.3±9.2 | 101.3±1.5 | -0.3±0.0 | -0.2±0.2 | 39.0±16.4 | 94.6±4.2 | **102.8±2.4** |
| | hammer (1%) | 90.5±7.8 | 90.6±3.5 | 93.0±15.8 | 0.2±0.0 | -0.0±0.1 | 70.0±12.6 | 97.8±12.3 | **113.8±1.6** |
| | relocate (1%) | 7.2±4.5 | 12.2±6.8 | 14.7±3.7 | -0.2±0.0 | -0.3±0.0 | 5.2±5.0 | 61.6±5.6 | **65.0±6.2** |
| | **Average** | 27.2 | 30.2 | 33.9 | 2.8 | 2.2 | 25.4 | 46.0 | **53.6** |
| Action | halfcheetah (10%) | 0.8±0.3 | 0.7±0.1 | 10.3±1.8 | 19.8±5.1 | 8.0±0.6 | 2.3±0.6 | 3.6±0.3 | **20.4±1.7** |
| | hopper (10%) | 18.3±2.4 | 17.7±2.3 | 32.8±3.9 | 1.8±0.0 | 33.0±4.3 | 26.3±2.8 | 32.1±6.0 | **37.6±5.7** |
| | walker2d (10%) | 5.1±1.4 | 6.1±3.5 | 13.9±9.6 | 3.1±2.4 | 6.4±0.5 | 9.8±1.9 | 22.2±4.0 | **30.4±2.5** |
| | kitchen-complete | 3.5±2.0 | 4.6±1.2 | 23.0±14.5 | 12.6±4.2 | 0.0±0.0 | 20.1±6.1 | 12.6±5.2 | **44.0±3.7** |
| | kitchen-partial | 29.9±3.0 | 34.0±3.5 | 5.9±2.5 | 5.4±6.3 | 0.0±0.0 | 33.4±5.6 | 26.2±0.8 | **40.8±3.7** |
| | kitchen-mixed | 34.9±3.8 | 39.5±4.5 | 17.0±7.7 | 0.0±0.0 | 0.0±0.0 | 41.0±10.4 | 30.4±2.2 | **44.4±3.7** |
| | door (1%) | 45.0±15.4 | 61.4±9.1 | 100.7±3.1 | -0.3±0.1 | -0.2±0.1 | 29.6±29.1 | 84.3±5.5 | **103.3±1.0** |
| | hammer (1%) | 76.4±15.2 | 67.8±6.9 | 99.7±17.8 | 0.2±0.0 | 0.1±0.1 | 68.7±29.4 | 72.0±5.8 | **116.3±0.8** |
| | relocate (1%) | 26.0±10.4 | 44.3±7.4 | 44.9±2.6 | -0.2±0.1 | -0.3±0.0 | 20.4±9.6 | 63.2±4.4 | **69.9±9.7** |
| | **Average** | 26.6 | 30.7 | 38.7 | 4.7 | 5.2 | 28.0 | 38.5 | **56.3** |
| Reward | halfcheetah (10%) | 2.4±0.2 | 2.9±0.3 | 14.3±3.6 | **30.4±1.9** | 2.2±0.7 | 6.1±1.2 | 9.3±0.9 | 21.7±4.0 |
| | hopper (10%) | 19.7±2.8 | 19.3±2.3 | 22.8±4.2 | 1.8±0.0 | 18.2±2.8 | 40.2±2.6 | 36.0±7.4 | **42.8±3.4** |
| | walker2d (10%) | 9.7±1.5 | 8.3±3.8 | 14.2±2.1 | 1.2±0.7 | 7.9±0.8 | 9.6±2.5 | 27.4±3.4 | **30.1±4.5** |
| | kitchen-complete | 36.0±11.5 | 38.2±8.4 | 43.2±5.7 | 1.4±2.4 | 0.0±0.0 | 52.8±6.8 | 43.9±4.3 | **65.6±4.4** |
| | kitchen-partial | 34.1±1.4 | 39.1±3.2 | 7.9±6.4 | 0.0±0.0 | 0.0±0.0 | 36.4±2.2 | 47.1±6.9 | **51.4±2.0** |
| | kitchen-mixed | 38.9±1.4 | 47.1±1.9 | 14.9±7.2 | 0.0±0.0 | 0.0±0.0 | 49.8±4.3 | 42.8±1.9 | 47.8±4.3 |
| | door (1%) | 76.0±5.9 | 75.0±3.2 | 102.3±2.7 | -0.3±0.0 | -0.2±0.0 | 56.1±9.5 | 99.0±2.3 | **103.7±0.8** |
| | hammer (1%) | 97.1±8.3 | 99.0±5.1 | 101.8±10.2 | 0.2±0.0 | 0.4±0.4 | 52.2±22.7 | 80.7±11.2 | **123.8±2.2** |
| | relocate (1%) | 36.1±8.6 | 32.2±6.7 | 53.1±3.1 | -0.3±0.1 | -0.3±0.0 | 9.1±5.3 | 71.8±7.7 | **84.9±1.6** |
| | **Average** | 38.9 | 40.1 | 41.6 | 3.8 | 3.1 | 34.7 | 50.9 | **63.5** |
| Average over all tasks | | 31.1 | 33.7 | 38.1 | 3.8 | 3.5 | 29.3 | 45.1 | **57.8** |

(and Adroit) tasks and conduct data corruption on the sampled data. We refrain from down-sampling the Kitchen dataset due to its already limited size. Detailed information about the dataset sizes can be found in Appendix C.4.

We compare RDT with several SOTA offline RL algorithms and corruption-robust methods, namely BC, RBC (Sasaki & Yamashina, 2020), DeFog (Hu et al., 2023), CQL (Kumar et al., 2020), UWMSG (Ye et al., 2024b), RIQL (Yang et al., 2024c), and DT (Chen et al., 2021). BC and RBC employ behaviour cloning loss within an MLP-based model for policy learning, while DeFog and DT utilize a Transformer architecture. We implement RBC with our Gaussian-weighted learning as an instance of (Sasaki & Yamashina, 2020). In Appendix D.1, we investigate the robustness of DT across various critical parameters. Drawing from these findings on DT's robustness, we establish a default implementation of DT, DeFog, RDT with a sequence length of 20 and a block number of 3. To ensure the validity of the findings, each experiment is repeated using 4 different random seeds, and we also report the standard variance over these random seeds.

## 4.2 EVALUATION UNDER VARIOUS DATA CORRUPTION

**Results under Random Corruption.** To address the first question, we first evaluate the RDT and baselines under the random data corruption scenario. The mean of normalized scores across different seeds is calculated as the evaluation criterion for each task. As shown in Table 1, RDT demonstrates superior performance in handling data corruption, achieving the highest score in 24 out of 27 settings. Notably, across all tasks, RDT consistently outperforms DT, with a significant overall improvement of 28.2%, underscoring its effectiveness in reducing DT's sensitivity to data corruption. Moreover, RBC slightly outperforms BC, further highlighting the effectiveness of the Gaussian weighted learning approach. However, prior offline RL methods, such as RIQL and UWMSG, fail to yield satisfactory results and even underperform BC, indicating temporal difference methods' weakness in the limited data domain. We further evaluate RDT under varying corruption scales and ratios in Appendix D.4, where RDT demonstrates superior robustness compared to other baselines.

Table 2: Results under adversarial data corruption. Each score is averaged across the task group.

| Attack | Task | BC | RBC | DeFog | CQL | UWMSG | RIQL | DT | **RDT** |
|---|---|---|---|---|---|---|---|---|---|
| State | MuJoCo (10%) | 9.9 | 9.4 | 12.9 | 4.2 | 9.0 | 11.0 | 21.9 | **23.6** |
| | KitChen | 23.4 | 28.9 | 20.0 | 2.2 | 0.0 | 40.4 | 37.9 | **49.1** |
| | Adroit (1%) | 60.2 | 53.6 | 75.6 | -0.1 | -0.1 | 44.5 | 85.3 | **95.4** |
| | **Average** | 31.2 | 30.6 | 36.2 | 2.1 | 2.9 | 32.0 | 48.4 | **56.0** |
| Action | MuJoCo (10%) | 4.2 | 4.3 | 10.3 | 7.3 | 16.9 | 7.1 | 13.4 | **21.6** |
| | KitChen | 6.2 | 11.5 | 3.9 | 0.9 | 0.0 | 8.0 | 5.4 | **34.0** |
| | Adroit (1%) | 8.4 | 20.7 | 51.8 | -0.1 | -0.1 | 42.4 | 47.4 | **80.3** |
| | **Average** | 6.3 | 12.2 | 22.0 | 2.7 | 5.6 | 19.2 | 22.1 | **45.3** |
| Reward | MuJoCo (10%) | 10.6 | 10.2 | 11.5 | 11.8 | 15.2 | 18.3 | 25.2 | **31.9** |
| | KitChen | 36.3 | 41.5 | 20.3 | 1.1 | 9.7 | 45.9 | 48.1 | **56.0** |
| | Adroit (1%) | 69.7 | 68.7 | 80.8 | -0.1 | 0.0 | 53.6 | 90.9 | **96.5** |
| | **Average** | 38.9 | 40.1 | 37.6 | 4.3 | -0.1 | 39.2 | 54.7 | **61.5** |
| Average over all tasks | | 25.5 | 27.7 | 31.9 | 3.0 | 4.4 | 30.1 | 41.7 | **54.3** |

**Results under Adversarial Corruption.** We further extend the analysis to examine the robustness of the RDT under an adversarial data corruption scenario. As illustrated in Table 2, we calculate the mean score across the task group. RDT consistently demonstrates robust performance, achieving the highest average scores under various attacked elements. Notably, RDT improves the average score by **105%** compared to DT in the adversarial action corruption scenario. Intriguingly, temporal-difference methods like CQL and UWMSG perform significantly worse than both BC and sequence modeling methods such as DT and DeFog, underscoring the potential of sequence modeling approaches. Detailed results for each task are provided in Appendix D for comprehensive analysis.

**Results under Mixed Corruption.** To present a more challenging scenario for assessing the robustness of RDT, we conduct experiments under mixed data corruption settings. In this setting, all three elements (states, actions, and rewards) are corrupted at a rate of $30\%$ and a scale of $1.0$. As shown in Figure 4, RDT consistently outperforms other baselines across all tasks, highlighting its superior stability even when faced with simultaneous and diverse data corruptions. Notably, in the challenging Kitchen and Adroit tasks with mixed adversarial settings, RDT surpasses DT and RIQL by an impressive margin of approximately 100%.

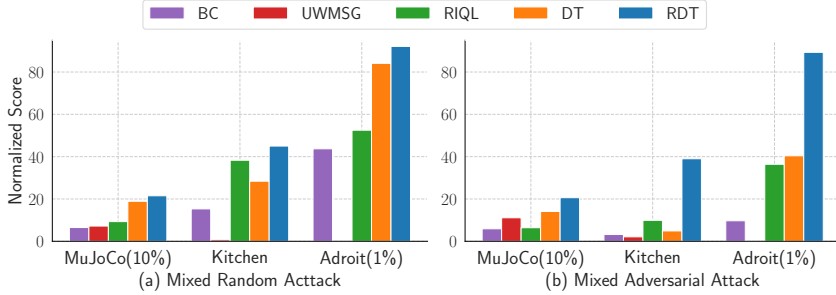

Figure 4: Results under (a) mixed random corruption and (b) mixed adversarial corruption.

### 4.3 Evaluation under Observation Perturbation during the Testing Phase

We investigate the robustness of RDT when deployed in perturbed environments after being trained on corrupted data, a challenging setting that includes both training-time and testing-time attacks. To address this, we evaluate RDT under two types of observation perturbations during the testing phase: **Random** and **Action Diff**, following prior works (Yang et al., 2022a; Zhang et al., 2020). Among these, **Action Diff** involves sampling multiple random noises to maximize the impact on the difference in policy output. The perturbation scale is used to control the extent of influence on the observation. Details of observation perturbations are provided in Appendix C.2.

The comparison results are presented in Figure 5, where all algorithms are trained on the offline dataset with mixed random corruption and evaluated under observation perturbation. These results

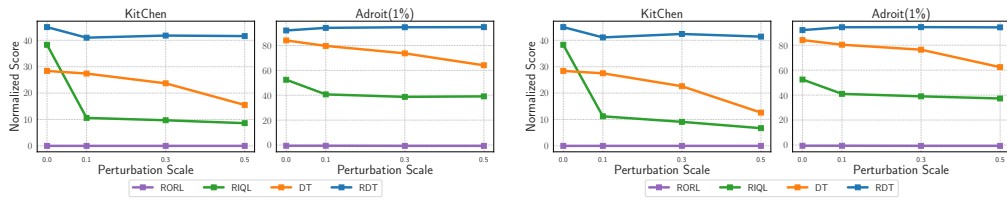

(a) Random observation perturbation.      (b) Action Diff observation perturbation.

Figure 5: Performance under various observation perturbation scales during the testing phase. All the algorithms are trained under **mixed random corruption** during the training phase.

demonstrate the superior robustness of RDT under the two types of observation perturbations, maintaining stability even at a high perturbation scale of 0.5. Notably, RORL (Yang et al., 2022a), a SOTA offline RL method designed to tackle testing-time observation perturbation, fails on these tasks due to data corruption. Additionally, DT and RIQL experience significant performance drops as the perturbation scale increases. Additional results comparing algorithms trained on offline datasets with corrupted states, actions, or rewards can be found in Appendix D.7.

## 4.4 ABLATION STUDY

We conduct comprehensive ablation studies to analyze the impact of each component on RDT's robustness. For evaluation, we use the "walker2d-medium-replay", "kitchen-compete", and "relocate-expert" datasets. Specifically, we compare several variants of RDT: (1) **DT(RP)**, which incorporates reward prediction in addition to the original DT; (2) **DT(RP) w. ED**, which adds embedding dropout to DT(RP); (3) **DT(RP) w. GWL**, which applies Gaussian weighted learning on top of DT(RP); and (4) **DT(RP) w. IDC**, which integrates only the iterative data correction method.

We evaluate the performance of these variants under different data corruption scenarios, as depicted in Figure 6. In summary, all variants demonstrate improvements over DT(RP), proving the effectiveness of the individual components. Notably, Gaussian weighted learning appears to provide the most significant contribution, particularly under reward attack. However, it is important to note that none of these tailored models outperform RDT, indicating that the integration of all proposed techniques is crucial for achieving optimal robustness. Additionally, while reward prediction could potentially lead to performance drops under reward attacks, it brings obvious improvements under action and state attacks for specific tasks. More ablation results about three individual component and reward prediction are provided in Appendix D.9 and Appendix D.2.

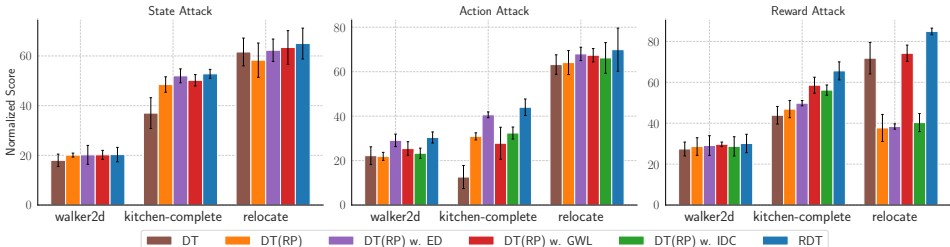

Figure 6: Ablation study on the impact of different proposed components of RDT.

## 4.5 IMPACT OF TRANSFORMER BACKBONE ON BASELINES

To study the impact of the transformer backbone on baselines, we adapted temporal difference learning baselines such as UWMSG and RIQL by incorporating the same transformer backbone used in DT and RDT. Specifically, we modified UWMSG and RIQL by equipping their policy networks with the transformer backbone while keeping the value networks unchanged. These modified algorithms are referred to as UWMSG w. TB and RIQL w. TB.

As shown in Figure 7, while the transformer backbone does not improve UWMSG, it significantly enhances RIQL's overall performance, particularly in the challenging Adroit task. The ineffectiveness of the sequence model on UWMSG is likely due to the inherent difficulties of Q-value learning.

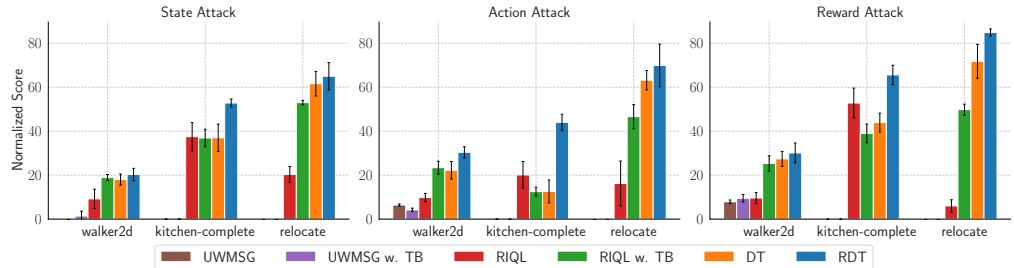

Figure 7: Ablation study on the impact of transformer backbone on baselines.

These comparative results suggest that merely adopting a transformer structure does not directly enable these baselines to outperform RDT; in some cases, it may even decrease performance.

## 5 RELATED WORK

**Robust Offline RL.** Several works have focused on testing-time robustness against environment shifts (Shi & Chi, 2022; Yang et al., 2022a; Panaganti et al., 2022; YANG & Xu, 2024). Regarding training-time robustness, Li et al. (2023) explores various types of reward attacks in offline RL and finds that certain biases can inadvertently enhance the robustness of offline RL methods to reward corruption. From a theoretical perspective, Zhang et al. (2022) propose a robust offline RL algorithm utilizing robust supervised learning oracles. Ye et al. (2024b) employ uncertainty weighting to address reward and dynamics corruption, providing theoretical guarantees. Ackermann et al. (2024) investigates a similar setting, focusing on trajectory-level non-stationarity in both rewards and dynamics. The most relevant research by Yang et al. (2024c) employs the Huber loss to handle heavy-tailedness and utilizes quantile estimators to balance penalization for corrupted data. Mandal et al. (2024); Liang et al. (2024) enhance the resilience of offline algorithms within the RLHF framework. It is important to note that these studies primarily focus on enhancing temporal difference methods, with no emphasis on leveraging sequence modeling techniques to tackle data corruption.

**Transformers for RL.** Recent research has redefined offline RL decision-making as a sequence modeling problem using Transformer architectures (Chen et al., 2021; Janner et al., 2021). A seminal work, Decision Transformer (DT) (Chen et al., 2021), uses trajectory sequences to predict subsequent actions. Trajectory Transformer (Janner et al., 2021) discretizes input sequences into tokens and employs beam search to predict the next action. These efforts have led to subsequent advancements. For instance, Prompt DT (Xu et al., 2022) integrates demonstrations for better generalization, while Xie et al. (2023) introduces pre-training with future trajectory information. Q-learning DT (Yamagata et al., 2023) refines the return-to-go using Q-values, while the Agentic Transformer (Liu & Abbeel, 2023) employs hindsight to relabel target returns. Additionally, several other works (Hu et al., 2024; Zhuang et al., 2024) utilize learned value functions to improve policy optimization. RiC (Yang et al., 2024b) extends DT to handle multiple objectives and applies it to large foundation models. LaMo (Shi et al., 2023) leverages pre-trained language models for offline RL, and DeFog (Hu et al., 2022) addresses robustness in specific frame-dropping scenarios. Our work deviates from these approaches by focusing on improving robustness against data corruption in offline RL.

## 6 CONCLUSION

In this study, we investigate the robustness of offline RL algorithms under various data corruption scenarios, with a specific focus on sequence modeling methods. Our empirical evidence suggests that current offline RL algorithms based on temporal difference learning are significantly susceptible to data corruption, especially in scenarios with limited data. To address this issue, we introduce the Robust Decision Transformer (RDT), a novel robust offline RL algorithm developed from the perspective of sequence modeling. Our comprehensive experiments highlight RDT's excellent robustness against various types of data corruption. Furthermore, we demonstrate RDT's superiority in handling both training-time and testing-time attacks. We hope that our findings will inspire further research into using sequence modeling methods to address data corruption challenges in increasingly complex and realistic scenarios.

ACKNOWLEDGMENTS

Baoxiang Wang is partially supported by the National Natural Science Foundation of China (62106213, 72394361) and an extended support project from the Shenzhen Science and Technology Program. Shuang Qiu acknowledges the support of GRF 16209124.

ETHICS STATEMENT

This research was conducted in compliance with all applicable ethical guidelines and institutional regulations. Since the study did not involve human participants, animals, or sensitive data, no specific ethical approvals were required. All data used in this research were obtained from publicly available sources, ensuring full transparency and reproducibility of the results.

REPRODUCIBILITY STATEMENT

We have taken several measures to ensure the reproducibility of our results. All datasets used in our experiments are publicly available and can be accessed through `https://github.com/Farama-Foundation/D4RL` and `https://github.com/polixir/NeoRL`. The implementation of all baselines and RDT is based on the codebase available at `https://github.com/tinkoff-ai/CORL`. Detailed instructions for setting up the environment and running the experiments are provided in Appendix C. Our implementation code is available at `https://github.com/jiawei415/RobustDecisionTransformer`. We encourage researchers to refer to Appendix C for more detailed information.

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

## CONTENTS

## A ALGORITHM PSEUDOCODE

To provide an overview and better understanding, we detail the implementation of the Robust Decision Transformer (RDT) in Algorithm 1.

---

**Algorithm 1** Robust Decision Transformer (RDT)

---

**Require:** Offline dataset $\mathcal{D}$, sequence model $\pi_\theta$, initialed mean $\mu_\delta$ and variance $\sigma_\delta^2$.
1: **for** training step= $1, 2, \ldots, T$ **do**
2:    Extract batch $\tau_{t:t+K-1}$ from the offline dataset $\mathcal{D}$.
3:    Update sequence model $\pi_\theta$ based on Eq. 6.
4:    Compute prediction errors $\delta_{a_t}$ and $\delta_{r_t}$ in batch data.
5:    Update corresponding mean $\mu_\delta$ and variance $\sigma_\delta^2$.
6: **end for**
7: **if** correction phase begins **then**
8:    Evaluate the $z$-score to identify the corrupted action $\hat{a}_t^{(i)}$ and reward $\hat{r}_t^{(i)}$.
9:    Substitute actions and rewards with the predicted actions and rewards in dataset $\mathcal{D}$.
10: **end if**

---

## B ADDITIONAL RELATED WORKS

**Offline RL.** Maintaining proximity between the policy and data distribution is essential in offline RL, as distributional shifts can lead to erroneous estimations (Levine et al., 2020). To counter this, offline RL algorithms are primarily divided into two categories. The first category focuses on policy constraints on the learned policy (Wang et al., 2018; Fujimoto et al., 2019; Li et al., 2020; Fujimoto & Gu, 2021; Kostrikov et al., 2021; Emmons et al., 2021; Yang et al., 2022b; 2023; Sun et al., 2024; Xu et al., 2023; Park et al., 2024). The other category learns pessimistic value functions to penalize OOD actions (Kumar et al., 2020; Yu et al., 2020; An et al., 2021; Bai et al., 2022; Yang et al., 2022a; Ghasemipour et al., 2022; Sun et al., 2022; Nikulin et al., 2023; Huang et al., 2024). To enhance the potential of offline RL in handling more complex tasks, recent research has integrated advanced techniques like GAN (Vuong et al., 2022; Wang et al., 2023), transformers (Chen et al., 2021; Chebotar et al., 2023; Yamagata et al., 2023) and diffusion models (Janner et al., 2022; Hansen-Estruch et al., 2023; Wang et al., 2022).

## C IMPLEMENTATION DETAILS

### C.1 DATA CORRUPTION DETAILS DURING TRAINING PHASE

Our study utilizes both random and adversarial corruption across three elements: states, actions, and rewards. We consider a range of tasks including MuJoCo, Kitchen, and Adroit. Particularly, we utilize the "medium-replay-v2" datasets in the MuJoCo tasks with sampling ratios of 10%, the "expert-v0" datasets in the Adroit tasks with a sampling ratio of 1%, and we employ full datasets for the tasks in the Kitchen due to their already limited data size. These datasets (Fu et al., 2020) are collected either during the training process of an SAC agent or from expert demonstrations, thereby providing highly diverse and representative tasks of the real world. To control the overall level of corruption within the datasets, we introduce two parameters $c$ and $\epsilon$ following previous work (Ye et al., 2024b; Yang et al., 2024c). The parameter $c$ signifies the rate of corrupted data within a dataset, whilst $\epsilon$ represents the scale of corruption observed across each dimension. We outline three types of random data corruption and present a comprehensive overview of a mixed corruption approach as follows. Note that in our setting, only three independent elements (i.e., states, actions, and rewards) are considered under the trajectory-based storage approach.

- **Random state attack**: We randomly sample $c \cdot N \cdot T$ states from all trajectories, where $N$ refer to the number of trajectories and $T$ represents the number of steps in a trajectory. We then modify the selected state to $\hat{s} = s + \lambda \cdot \text{std}(s), \lambda \sim \text{Uniform}[-\epsilon, \epsilon]^{d_s}$. Here, $d_s$ represents the dimension of states, and "std$(s)$" is the $d_s$-dimensional standard deviation of

all states in the offline dataset. The noise is scaled according to the standard deviation of each dimension and is independently added to each respective dimension.

- **Random action attack**: We randomly select $c \cdot N \cdot T$ actions from all given trajectories, and modify the action to $\hat{a} = a + \lambda \cdot \text{std}(a), \lambda \sim \text{Uniform}[-\epsilon, \epsilon]^{d_a}$, where $d_a$ represents the dimension of actions and "std$(a)$" is the $d_a$-dimensional standard deviation of all actions in the offline dataset.

- **Random reward attack**: We randomly sample $c \cdot N \cdot T$ rewards from all given trajectories, and modify the reward to $\hat{r} \sim \text{Uniform}[-30 \cdot \epsilon, 30 \cdot \epsilon]$. We multiply by 30 because we have noticed that offline RL algorithms tend to be resilient to small-scale random reward corruption (also observed in Li et al. (2023)), but would fail when faced with large-scale random reward corruption.

In addition, three types of adversarial data corruption are detailed as follows:

- **Adversarial state attack**: We first pretrain IQL agents with a Q function $Q_p$ and policy function $\pi_p$ on clean datasets. Then, we randomly sample $c \cdot N \cdot T$ states, and modify them to $\hat{s} = \min_{\hat{s} \in \mathbb{B}_d(s, \epsilon)} Q_p(\hat{s}, a)$. Here, $\mathbb{B}_d(s, \epsilon) = \{\hat{s} \| |\hat{s} - s| \leq \epsilon \cdot \text{std}(s)\}$ regularizes the maximum difference for each state dimension. The optimization is implemented through Projected Gradient Descent similar to prior works (Madry et al., 2017; Zhang et al., 2020; Yang et al., 2024c). Specifically, We first initialize a learnable vector $z \in [-\epsilon, \epsilon]^{d_s}$, and then conduct a 100-step gradient descent with a step size of 0.01 for $\hat{s} = s + z \cdot \text{std}(s)$, and clip each dimension of $z$ within the range $[-\epsilon, \epsilon]$ after each update.

- **Adversarial action attack**: We use the pretrained IQL agent with a Q function $Q_p$ and a policy function $\pi_p$. Then, we randomly sample $c \cdot N \cdot T$ actions, and modify them to $\hat{a} = \min_{\hat{a} \in \mathbb{B}_d(a, \epsilon)} Q_p(s, \hat{a})$. Here, $\mathbb{B}_d(a, \epsilon) = \{\hat{a} \| |\hat{a} - a| \leq \epsilon \cdot \text{std}(a)\}$ regularizes the maximum difference for each action dimension. The optimization is implemented through Projected Gradient Descent, as discussed above.

- **Adversarial reward attack**: We randomly sample $c \cdot N \cdot T$ rewards, and directly modify them to: $\hat{r} = -\epsilon \times r$.

## C.2 OBSERVATION PERTURBATION DETAILS DURING THE TESTING PHASE

We evaluate RDT and other baselines under two types of observation perturbations during the testing phase: **Random** and **Action Diff** perturbations, as described in prior works (Yang et al., 2022a; Zhang et al., 2020). Generally, offline RL algorithms are sensitive to observation perturbations, often resulting in a significant performance drop. The detailed implementations are as follows:

- **Random**: We sample perturbed states within an $l_\infty$ ball of norm $\epsilon$ centering at the original state. Specifically, we create the perturbation set $\mathbb{B}_d(s, \epsilon) = \{\hat{s} : d(s, \hat{s}) \leq \epsilon\}$ for state $s$, where $d(\cdot)$ is the $l_\infty$ norm, and sample one perturbed state to return to the agent.

- **Action Diff**: This is an adversarial attack based on a pretrained IQL deterministic policy $\mu(s)$. We first sample 50 perturbed states within an $l_\infty$ ball of norm $\epsilon$ and then find the one that maximizes the difference in actions: $\max_{\hat{s} \in \mathbb{B}_d(s, \epsilon)} ||\mu(s) - \mu(\hat{s})||^2$.

The parameter $\epsilon$ controls the scale of observation perturbations. We consider $\{0.0, 0.1, 0.3, 0.5\}$ for $\epsilon$ in our experiments. In this setup, we first train offline RL algorithms under various data corruption settings, and then evaluate their performance in environments with observation perturbations.

## C.3 IMPLEMENTATION DETAILS OF RDT

We implement RDT and other baselines using the existing code base[1]. Specifically. we build the network with 3 Transformer blocks, incorporating one MLP embedding layer for each key element: state, action, and return-to-go. We update the neural network using the AdamW optimizer, with a learning rate set at $1 \times 10^{-4}$ and a weight decay of $1 \times 10^{-4}$. The batch size is set to 64, with a sequence length of 20, to ensure effective and efficient training. To maintain stability during training,

---

[1] https://github.com/tinkoff-ai/CORL

we adopt the state normalization as in Yang et al. (2024c). During the training phase, we train DeFog, DT and RDT for 100 epochs and other baselines with 1000 epochs. Each epoch is characterized by 1000 gradient steps. For evaluative purposes, we rollout each agent in environment across 10 trajectories, each with a maximum length of 1000, and we average the returns from these rollouts for comparison. We ensure the consistency and reliability of our reported results by averaging them over 4 unique random seeds. All experiments are conducted on P40 GPUs.

As for the hyperparameters of RDT, we configure the values for the embedding dropout probability as $p$, Gaussian Weighted coefficient $\beta_a$ and $\beta_r$, and iterative data correction threshold $\zeta$ through the sweeping mechanism. We start the iterative data correction at the 50th epoch for all tasks. We use iterative action correction exclusively for action corruption and iterative reward correction exclusively for reward corruption; therefore, only a single $\zeta$ value is shown in Table 3. **It's noteworthy that we do not conduct a hyperparameter search for individual datasets; instead, we select hyperparameters that apply to the entire task group.** As such, RDT doesn't require extensive hyperparameter tuning as previous methods. The exact hyperparameters used for the random corruption experiment are detailed in Table 3. We employ almost same hyperparameters for the adversarial corruption experiment, further attesting to the robustness and stability of RDT.

Table 3: Hyperparameters used for RDT under the random corruption.

| Tasks | Attack Element | $p$ | $(\beta_a, \beta_r)$ | $\zeta$ |
|---|---|---|---|---|
| MuJoCo (10%) | state | 0.2 | (1.0, 1.0) | – |
| | action | 0.2 | (1.0, 1.0) | 6.0 |
| | reward | 0.2 | (1.0, 1.0) | 6.0 |
| Kitchen | state | 0.1 | (30.0, 30.0) | – |
| | action | 0.1 | (30.0, 30.0) | 5.0 |
| | reward | 0.1 | (30.0, 30.0) | 6.0 |
| Adroit (1%) | state | 0.1 | (0.1, 0.0) | – |
| | action | 0.1 | (10.0, 0.0) | 6.0 |
| | reward | 0.1 | (0.1, 0.1) | 6.0 |

### C.4 DETAILED INFORMATION ON DATASET

Since we conduct experiments on both the full and down-sampled datasets, we provide detailed information about the number of transitions and trajectories. As seen from Table 4, after down-sampling, the number of transitions remains of the same order of magnitude for most datasets, except for the Kitchen. Due to the small number of transitions in the original full "kitchen-complete" dataset, we opted not to perform down-sampling on the Kitchen.

| Dataset | halfcheetah | hopper | walker2d |
|---|---|---|---|
| # Transitions | 202000 | 402000 | 302000 |
| # Trajectories | 202 | 2041 | 1093 |

| Dataset | halfcheetah(10%) | hopper(10%) | walker2d(10%) |
|---|---|---|---|
| # Transitions | 20000 | 32921 | 27937 |
| # Trajectories | 20 | 204 | 109 |

| Dataset | kitchen-complete | kitchen-partial | kitchen-mixed |
|---|---|---|---|
| # Transitions | 3680 | 136950 | 136950 |
| # Trajectories | 19 | 613 | 613 |

| Dataset | door(1%) | hammer(1%) | relocate(1%) |
|---|---|---|---|
| # Transitions | 10000 | 10000 | 10000 |
| # Trajectories | 50 | 50 | 50 |

Table 4: Detailed information about the number of transitions and trajectories.

# D  ADDITIONAL EXPERIMENTS

To provide a comprehensive understanding of our work, we conduct several additional analyses. First, we investigate the key hyperparameters that influence the robustness of DT. Next, we discuss the impact of reward prediction on DT. Following this, we demonstrate how these hyperparameters affect RDT and assess the robustness of RDT under various scales of data corruption. We then compare the training time between RDT and baseline models. Finally, we present detailed results of RDT under both training-time corruption and testing-time perturbation.

## D.1  IMPACT OF DECISION TRANSFORMER'S HYPERPARAMETERS

We aim to understand the robustness characteristic of sequence modeling. The key feature of the DT is its utilization of sequence modeling, with the incorporation of the Transformer model. Therefore, we identify two crucial factors that may influence the performance of DT in the context of sequence modeling: the model structure and the input data.

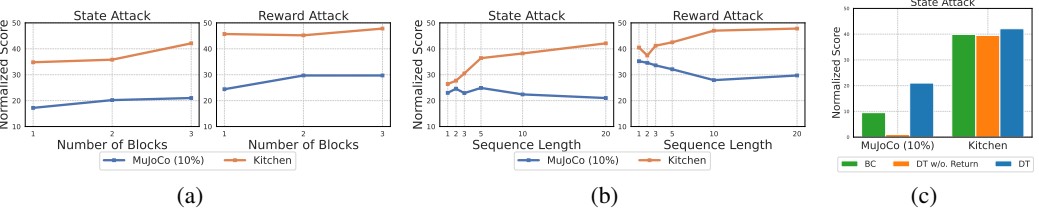

Figure 8: Ablation study on the robustness of DT. (a) Analyzing the influence of Transformer block number while keeping sequence length with 20. (b) The effect of sequence length while maintaining 3 Transformer blocks. (c) Investigation on how the return-to-go impacts DT. We create the variant **DT w/o. Return**, which excludes the use of return-to-go as inputs.

We explore the influence of the model's structure by modifying the number of Transformer blocks in DT. As depicted in Figure 8(a), the performance on both MuJoCo and Kitchen tasks consistently improves with increased number of blocks. Therefore, the enhancement can be partially attributed to the increased capabilities of the larger model.

We also examine the impact of the inputs for the sequence model. There are two primary differences between the input of DT and that of conventional BC. DT predicts actions based on historical data and return-to-go elements, whereas BC relies solely on the current states. To assess the influence of these factors, we independently evaluate the impact of the length of history and return-to-go elements. The results in Figure 8(b) and (c) reflect that Kitchen tasks favour longer historical input and are less reliant on return-to-go elements. Conversely, MuJoCo tasks demonstrated a different trend. This discrepancy could be due to the sparse reward structure of Kitchen tasks, where return-to-go elements do not provide sufficient information, leading the agent to rely more heavily on historical data for policy learning. In contrast, MuJoCo tasks, with their denser reward structures, rely more on return-to-go elements for optimal policy learning.

In summary, the robustness of DT can be attributed to the model capacity, history length, and return-to-go conditioning. Based on these findings on the robustness of DT, we adopt a sequence length of 20 and a block number of 3 as the default implementation for all sequence modeling methods, including DT, DeFog, and RDT.

## D.2  ABLATION STUDY ON REWARD PREDICTION

The nature of the sequence model allows us to predict not only actions but also other elements, such as state and return-to-go. Previous studies have shown that predicting these additional elements does not significantly enhance performance Chen et al. (2021). However, we find that predicting rewards can provide advantages in the context of data corruption. We introduce **DT(RP)**, a variant of the original DT, which predicts rewards in addition to actions. As illustrated in Figure 9, predicting rewards indeed improves performance under random state and action corruption conditions. How-

ever, **DT(RP)** performs poorly under reward corruption due to the degraded quality of reward labels. This issue can be mitigated via embedding dropout, Gaussian weighted learning, and iterative data correction in RDT.

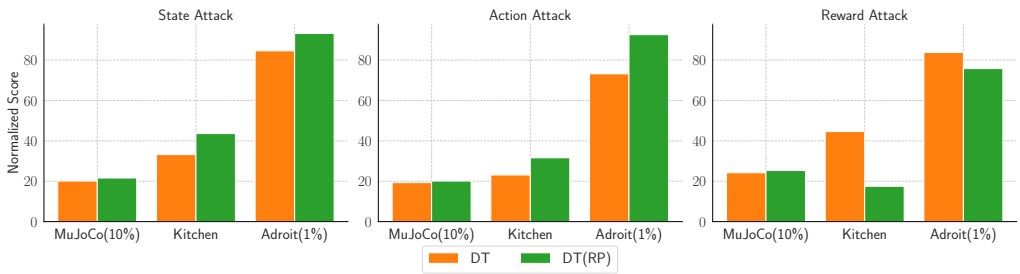

Figure 9: Ablation study on the impact of reward prediction.

### D.3 ABLATION STUDY ON HYPERPARAMETERS OF RDT

We conduct experiments to investigate the impact on hyperparameters of RDT under scenarios of random action corruption.

In the study of the embedding dropout technique, depicted in Figure 10(a), we discover that applying a moderate dropout probability (e.g., 0.1) to embeddings enhances resilience against data corruption. However, a larger dropout rate ($\geq 0.5$) can lead to performance degradation.

As shown in Figure 10(b), regarding the Gaussian weighted coefficient $\beta_a$, the *walker2d* task favours lower values of $\beta_a$ such as 0.1, while the *kitchen-complete* and *relocate* tasks favour larger $\beta_a$ values, suggesting that these tasks are less impacted by Gaussian weight learning.

In terms of the iterative data correction threshold $\zeta$, the results are highlighted inFigure 10(c). The *walker2d* and *kitchen-complete* tasks favour lower values of $\zeta$, whereas the *relocate* task exhibits the opposite preference. Notably, in the *relocate* task, RDT shows markedly enhanced performance after correcting outliers (with $\zeta \geq 5$) in the dataset.

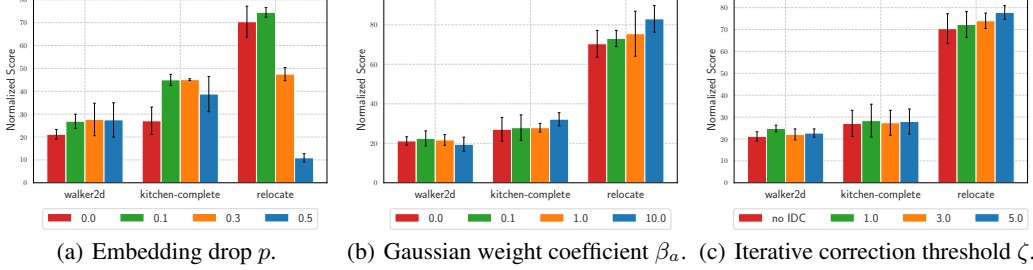

(a) Embedding drop $p$.  (b) Gaussian weight coefficient $\beta_a$.  (c) Iterative correction threshold $\zeta$.

Figure 10: Ablation study on the impact of hyperparameters under action corruption scenario.

### D.4 VARYING CORRUPTION RATES AND SCALES

We have evaluated the effectiveness of RDT under a data corruption rate of $0.3$ and a scale of $1.0$ in Section 4.2. We further examine the robustness of RDT under different corruption rates from $\{0.0, 0.1, 0.3, 0.5\}$ and scales from $\{0.0, 1.0, 2.0\}$. As illustrated in Figure 11, RDT consistently delivers superior performance compared to other baselines across different corruption rates and scales. However, we also observed a decline in RDT's performance when it encountered state corruption with high rates. This decline can be attributed to the distortion of the state's distribution caused by the high corruption rates, resulting in a significant deviation from the clean test environment to the corrupted datasets. Enhancing the robustness of RDT against state corruption, especially under high corruption rates and scales, holds potential for future advancements.

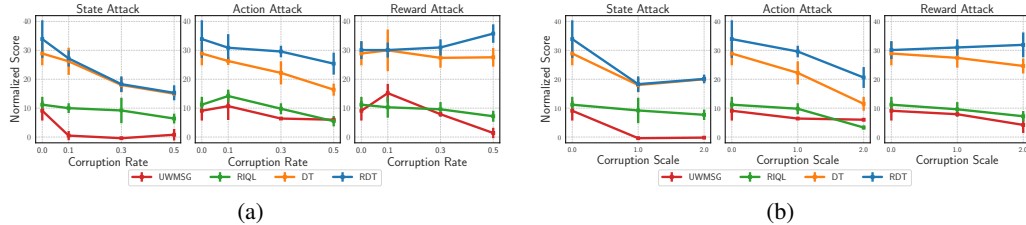

Figure 11: Results under various corruption rates (a) and scales (b) on the "walker2d" task.

## D.5 DETAILED RESULTS FOR THE EXPERIMENTS ON ADVERSARIAL ATTACK

The detailed outcomes of each task under adversarial data corruption are presented in Table 5. Notably, RDT outperforms baseline models in the majority of tasks, **achieving the best average performance**. In particular, RDT consistently outperforms DT, demonstrating the effectiveness of our three robust techniques.

Table 5: Detailed comparative results under adversarial data corruption.

| Attack | Task | BC | RBC | DeFog | CQL | UWMSG | RIQL | DT | **RDT** |
|---|---|---|---|---|---|---|---|---|---|
| State | halfcheetah (10%) | 2.1±0.6 | 2.4±0.3 | 4.8±0.7 | **10.0**±1.6 | 7.2±2.6 | 3.6±0.5 | 5.4±0.7 | 7.5±0.9 |
| | hopper (10%) | 19.1±3.2 | 18.7±4.4 | 21.8±7.4 | 1.8±0.8 | 14.7±3.4 | 19.1±6.9 | 38.9±2.9 | **39.1±4.6** |
| | walker2d (10%) | 8.7±1.8 | 7.1±0.2 | 12.0±3.6 | 0.7±1.5 | 5.1±2.3 | 10.4±1.9 | 21.3±1.5 | **24.2±3.2** |
| | kitchen-complete | 18.9±7.0 | 20.8±9.9 | 43.5±5.5 | 4.6±4.8 | 0.0±0.0 | 51.9±3.3 | 37.6±3.2 | **61.1**±2.0 |
| | kitchen-partial | 27.1±4.2 | 30.8±3.8 | 8.9±5.2 | 0.0±0.0 | 0.0±0.0 | 35.4±5.8 | 37.9±6.2 | **41.6±5.1** |
| | kitchen-mixed | 24.1±3.1 | 35.2±3.7 | 7.6±6.3 | 2.0±3.5 | 0.0±0.0 | 33.9±11.2 | 38.1±6.0 | **44.5±5.1** |
| | door (1%) | 71.1±8.8 | 66.1±10.2 | 102.2±1.4 | -0.3±0.0 | -0.3±0.1 | 47.3±24.8 | 98.2±4.8 | **104.0**±1.0 |
| | hammer (1%) | 82.5±11.1 | 79.3±8.4 | 102.7±8.8 | 0.2±0.0 | 0.1±0.1 | 69.1±23.4 | 90.1±11.9 | **110.9**±9.0 |
| | relocate (1%) | 27.0±2.6 | 15.5±7.2 | 22.1±3.5 | -0.3±0.0 | -0.3±0.0 | 17.1±9.5 | 67.6±2.6 | **71.3**±2.5 |
| | **Average** | 31.2 | 30.6 | 36.2 | 2.1 | 2.9 | 32.0 | 48.4 | **56.0** |
| Action | halfcheetah (10%) | -0.1±0.1 | -0.0±0.2 | 8.9±1.0 | **19.0**±2.2 | 6.7±1.1 | 0.8±0.4 | 2.4±0.1 | 17.1±3.4 |
| | hopper (10%) | 10.9±3.7 | 11.1±2.2 | 16.1±10.4 | 1.5±0.5 | **37.4**±4.0 | 17.1±1.4 | 28.8±3.3 | 31.6±1.8 |
| | walker2d (10%) | 1.9±0.7 | 1.9±0.5 | 5.8±2.4 | 1.4±1.0 | 6.7±1.0 | 3.3±1.2 | 9.0±1.4 | **16.0**±1.7 |
| | kitchen-complete | 6.6±0.6 | 5.6±3.0 | 11.5±11.1 | 1.8±1.9 | 0.0±0.0 | 20.6±4.5 | 11.5±2.3 | **25.5**±1.7 |
| | kitchen-partial | 5.8±3.6 | 6.9±2.3 | 0.0±0.0 | 1.0±1.7 | 0.0±0.0 | 0.0±0.0 | 1.9±1.4 | **30.8**±4.5 |
| | kitchen-mixed | 6.4±7.4 | 22.1±4.3 | 0.1±0.2 | 0.0±0.0 | 0.0±0.0 | 3.4±4.5 | 2.8±3.4 | **45.9**±4.6 |
| | door (1%) | 6.0±1.3 | 11.8±5.7 | **102.8**±1.1 | -0.3±0.1 | -0.2±0.1 | 38.7±21.0 | 62.2±7.4 | 97.6±5.5 |
| | hammer (1%) | 16.3±10.2 | 44.0±12.2 | 37.9±9.9 | 0.2±0.0 | 0.2±0.1 | 88.1±14.6 | 62.2±16.7 | **107.7**±1.3 |
| | relocate (1%) | 2.7±2.7 | 6.2±3.4 | 14.7±4.7 | -0.2±0.1 | -0.3±0.0 | 0.4±0.5 | 18.0±6.0 | **35.7**±3.1 |
| | **Average** | 6.3 | 12.2 | 22.0 | 2.7 | 5.6 | 19.2 | 22.1 | **45.3** |
| Reward | halfcheetah (10%) | 2.4±0.2 | 2.9±1.0 | 15.0±2.2 | **31.2**±1.6 | 1.9±0.2 | 10.9±1.6 | 9.5±0.6 | 23.3±5.5 |
| | hopper (10%) | 19.7±2.8 | 19.3±3.1 | 15.6±8.4 | 1.8±0.0 | 28.6±9.9 | 34.5±5.1 | 37.2±6.5 | **37.2**±6.6 |
| | walker2d (10%) | 9.7±1.5 | 8.3±1.7 | 4.0±1.9 | 2.2±2.4 | 9.7±3.1 | 9.3±1.2 | 29.0±3.6 | **35.2**±2.9 |
| | kitchen-complete | 36.0±11.5 | 38.2±5.0 | 48.0±2.4 | 3.4±5.8 | 0.0±0.0 | 51.5±2.6 | 45.0±5.0 | **65.9**±2.0 |
| | kitchen-partial | 34.1±1.4 | 39.1±1.8 | 9.5±7.9 | 0.0±0.0 | 0.0±0.0 | 37.2±8.5 | 45.1±6.1 | **46.1**±1.6 |
| | kitchen-mixed | 38.9±1.4 | 47.1±2.0 | 3.5±3.3 | 0.0±0.0 | 0.0±0.0 | 48.9±4.7 | 54.1±2.3 | **56.1**±2.8 |
| | door (1%) | 76.0±5.9 | 75.0±9.0 | **102.2**±1.0 | -0.3±0.1 | -0.2±0.1 | 73.4±11.3 | 98.9±2.8 | 101.4±0.6 |
| | hammer (1%) | 97.1±8.3 | 99.0±11.8 | 92.1±14.0 | 0.2±0.0 | 0.1±0.1 | 69.1±27.7 | 97.1±11.3 | **110.9**±1.6 |
| | relocate (1%) | 36.1±8.6 | 32.2±3.2 | 48.1±5.0 | -0.3±0.1 | -0.3±0.0 | 18.1±6.0 | 76.8±9.1 | **77.2**±4.2 |
| | **Average** | 38.9 | 40.1 | 37.6 | 4.3 | 4.4 | 39.2 | 54.7 | **61.5** |
| Average over all tasks | | 25.5 | 27.7 | 31.9 | 3.0 | 4.3 | 30.1 | 41.7 | **54.3** |

## D.6 COMPARISON RESULTS ON MUJOCO TASKS WITH FULL DATASET

We further evaluate the robustness of RDT on MuJoCo tasks using the full dataset. As shown in Table 6, RIQL is comparable to DT in this setting, while RDT achieves the highest overall average score. These findings further demonstrate the effectiveness of our proposed robust techniques across different dataset scales.

Our reported results for RIQL on MuJoCo tasks using the full 100% dataset show some discrepancies compared to the original paper (Yang et al., 2024c). To investigate this, we plotted the learning curve of RIQL under an action attack, as shown in Figure 12. In comparison to Figure 17 in the RIQL paper, we found that RIQL may require more training epochs to converge when using the

Table 6: Results under random data corruption on MuJoCo tasks with 100% dataset.

| Attack | Task | BC | RBC | DeFog | CQL | UWMSG | RIQL | DT | **RDT** |
|---|---|---|---|---|---|---|---|---|---|
| State | halfcheetah | 32.4±1.9 | **33.0**±0.9 | 14.9±7.1 | 23.7±4.4 | 2.2±0.6 | 19.9±2.1 | 27.5±2.5 | 30.8±1.8 |
| | hopper | 27.1±8.6 | 26.6±2.9 | 21.1±6.8 | 34.4±22.8 | 21.6±8.0 | 34.0±13.4 | 51.3±14.0 | **56.6**±2.9 |
| | walker2d | 19.2±4.7 | 16.8±6.0 | 17.4±2.4 | 33.7±10.8 | 0.8±1.7 | 14.2±1.2 | 47.6±4.9 | **53.4**±4.0 |
| | **Average** | 26.2 | 25.4 | 17.8 | 30.6 | 8.2 | 22.7 | 42.1 | **46.9** |
| Action | halfcheetah | 34.2±1.4 | 34.6±1.9 | 34.2±1.8 | 43.9±0.4 | **49.9**±1.2 | 41.3±1.4 | 35.1±2.4 | 37.4±1.4 |
| | hopper | 22.5±4.8 | 28.8±10.1 | 40.8±11.6 | 29.8±8.8 | 39.0±6.8 | 59.6±12.7 | 63.2±5.6 | **63.2**±5.8 |
| | walker2d | 18.3±2.6 | 21.3±3.6 | 50.7±6.0 | 1.8±3.6 | 53.3±16.8 | **72.6**±17.3 | 59.0±5.5 | 61.4±1.7 |
| | **Average** | 25.0 | 28.2 | 41.9 | 25.2 | 47.4 | **57.8** | 52.4 | 54.0 |
| Reward | halfcheetah | 35.9±0.5 | 34.7±1.4 | 32.6±2.3 | **43.3**±0.3 | 37.2±5.8 | 42.7±0.9 | 38.4±0.3 | 39.6±0.8 |
| | hopper | 26.6±6.9 | 29.5±8.6 | 29.2±18.3 | 29.9±7.2 | 65.4±26.0 | 59.6±4.8 | 54.3±10.7 | **70.9**±8.7 |
| | walker2d | 23.5±9.8 | 22.9±6.9 | 40.1±9.4 | 38.3±25.9 | 51.3±10.6 | **79.0**±5.8 | 64.2±4.0 | 66.7±4.0 |
| | **Average** | 28.7 | 29.0 | 34.0 | 37.2 | 51.3 | **60.4** | 52.3 | 59.0 |
| Average over all tasks | | 26.6 | 27.6 | 31.2 | 31.0 | 35.6 | 47.0 | 48.9 | **53.3** |

full dataset, which is reasonable. Given our focus on the limited data setting (10% for MuJoCo), we observed that 1,000 epochs are sufficient for both RDT and baselines like RIQL to converge, as detailed in Appendix D.8. Therefore, we use 1,000 training epochs by default, which, however, may not be optimal for the full dataset setting.

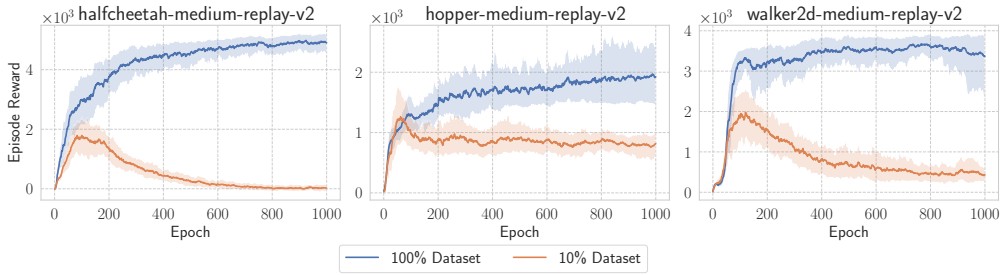

Figure 12: Learning curve of RIQL under action attack.

## D.7 ADDITIONAL RESULTS UNDER OBSERVATION PERTURBATION DURING TESTING PHASE

To further demonstrate the robustness of RDT, we conduct additional evaluations under observation perturbations during the testing phase. Specifically, we compare different algorithms trained with offline datasets under conditions of random state, action, or reward corruption in environments with observation perturbations. As shown in Figures 13 to 15, RDT consistently maintains superior performance and robustness across observation perturbations in both the Kitchen and Adroit tasks. We leave the investigation of robustness improvements over MuJoCo tasks for future work.

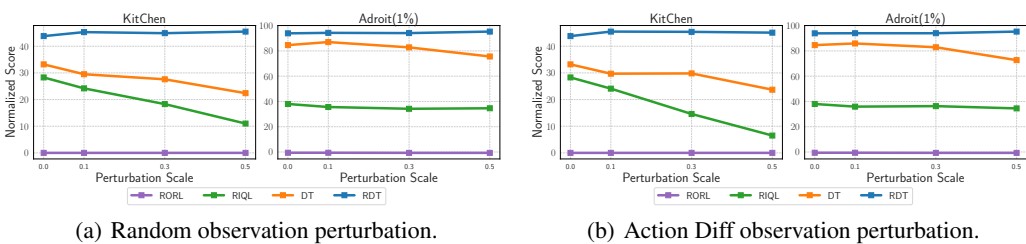

(a) Random observation perturbation.  (b) Action Diff observation perturbation.

Figure 13: Performance under varying observation perturbation scales during the testing phase. All the algorithms are trained under **random state corruption** during the training phase.

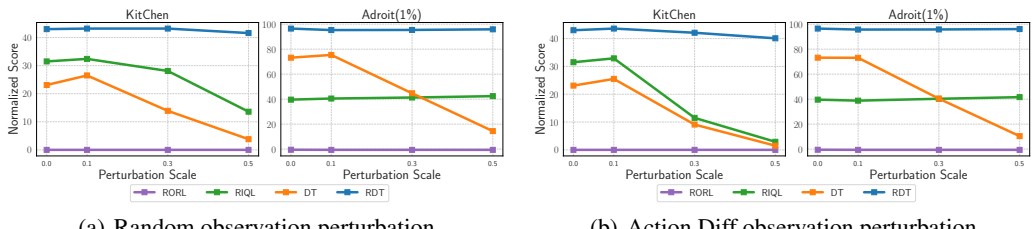

(a) Random observation perturbation.

(b) Action Diff observation perturbation.

Figure 14: Performance under varying observation perturbation scales during the testing phase. All the algorithms are trained under **random action corruption** during the training phase.

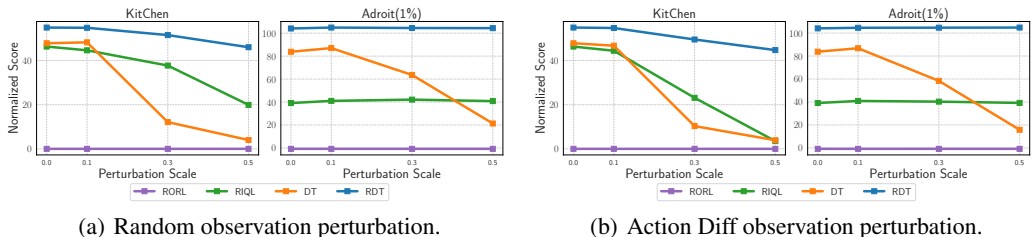

(a) Random observation perturbation.

(b) Action Diff observation perturbation.

Figure 15: Performance under varying observation perturbation scales during the testing phase. All the algorithms are trained under **random reward corruption** during the training phase.

## D.8 TRAINING TIME

We train all algorithms on the same P40 GPU, and the training times are presented in Table 7. As expected, BC has the shortest epoch time due to its simplicity. Notably, RDT requires only slightly more time per epoch than DT because it incorporates three robust techniques. Although RIQL requires less time per epoch than RDT, it necessitates 10 times more epochs to converge and still does not achieve satisfactory performance in the limited data regime. In conclusion, RDT achieves superior performance and robustness without imposing significant computational costs in terms of total training time.

Table 7: Training time on the "walker2d-medium-replay-v2" dataset.

| Method | BC | CQL | UWMSG | RIQL | DT | **RDT** |
|---|---|---|---|---|---|---|
| Epoch Num | 1000 | 1000 | 1000 | 1000 | 100 | 100 |
| Epoch Time (s) | 3.7 | 33.8 | 22.2 | 14.9 | 44.0 | 46.1 |
| Total Time (h) | 1.03 | 9.39 | 6.17 | 4.14 | 1.22 | 1.28 |

## D.9 ADDITIONAL ABLATION STUDY ON THREE ROBUST TECHNIQUES

We investigate the impact of each individual technique in Section 4.4 and demonstrate that integrating all techniques is essential for achieving optimal robustness. To provide a more comprehensive analysis of our proposed techniques, we create the variants RDT w/o ED, RDT w/o GWL, and RDT w/o IDC, which eliminate the components Embedding Dropout, Gaussian Weighted Learning, and Iterative Data Correction from RDT, respectively. As illustrated in Figure 16, the results are consistent with the findings in Section 4.4. This consistency reinforces our initial conclusion that the integration of all proposed techniques is crucial for achieving optimal robustness.

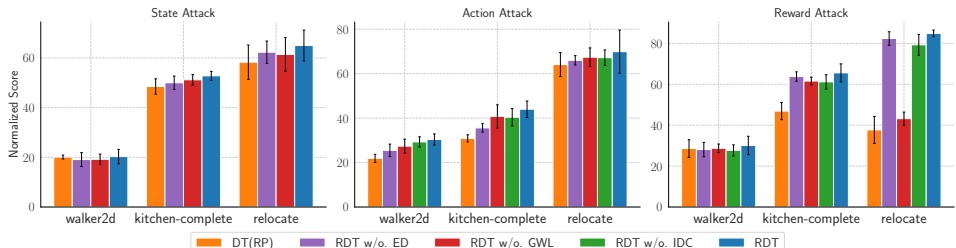

Figure 16: Additional ablation study on the impact of proposed techniques.

## D.10 EVALUATION UNDER VARIOUS DATASET SCALES

We conducted experiments across various dataset scales to understand the impact of dataset scales. Specifically, we evaluate RDT on the MuJoCo tasks with 20% and 50% of the "medium-replay-v2" dataset and the Adroit tasks with 5% and 10% of the "expert-v0" dataset. The comparison results are shown in Figure 17. Our empirical findings indicate that: (1) The overall performance of all methods improves as the dataset size increases. (2) Moreover, temporal-difference-based methods like RIQL are comparable to vanilla sequence modeling methods like DT when the dataset is large, but they are less preferable when the dataset size is limited. (3) Notably, RDT consistently outperforms other baselines across all dataset sizes, validating the effectiveness and importance of our proposed robustness techniques.

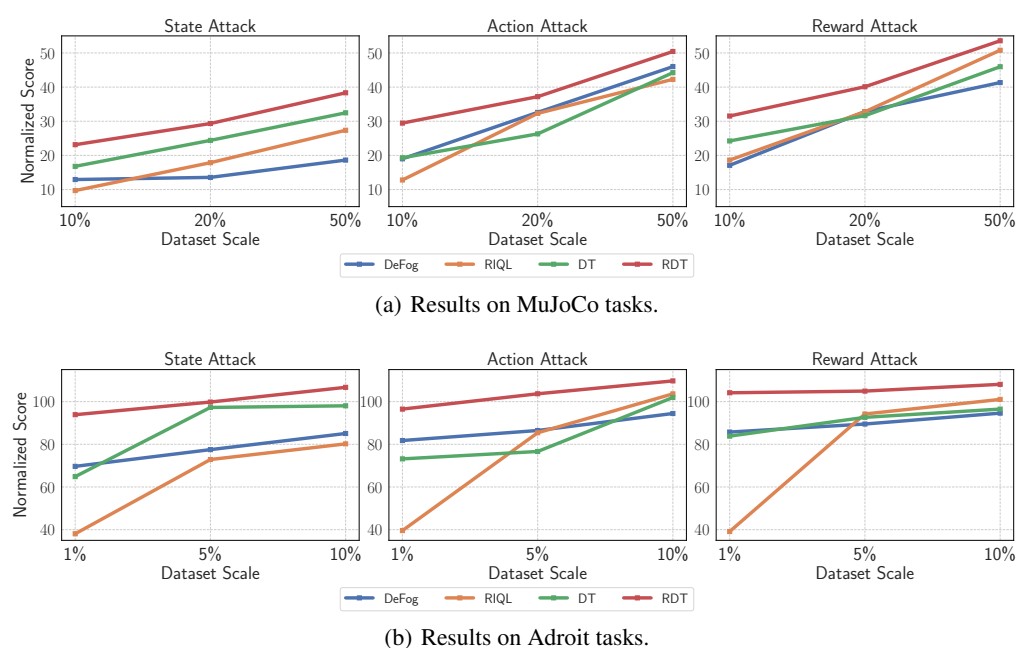

Figure 17: Performance of different algorithms across various dataset scales.

## D.11 Performance under MuJoCo Dataset with Different Quality Levels

To further validate the robustness of the RDT across varying dataset quality levels, we conduct evaluations using both the "expert-v2" and "medium-v2" datasets on MuJoCo tasks. We randomly sample 2% for both datasets, respectively, resulting in both datasets having a similar size of approximately $2 \times 10^4$ transitions. We include RIQL, DeFog, and DT as baselines, and the results are presented in Tables 8 and 9. Notably, RDT consistently outperforms the other baselines across datasets of different quality levels. This superior performance reinforces RDT's capability to efficiently handle a range of dataset qualities.

Table 8: Results on "medium-v2" dataset under random data corruption.

| Attack | Task | DeFog | RIQL | DT | **RDT** |
|---|---|---|---|---|---|
| State | halfcheetah (2%) | 7.5±1.4 | 18.0±1.5 | 15.7±1.3 | **22.3**±1.1 |
| | hopper (2%) | 20.4±8.3 | 45.7±7.3 | 48.6±3.2 | **52.2**±5.7 |
| | walker2d (2%) | 14.2±1.6 | 25.4±5.0 | 20.5±6.0 | **28.0**±8.0 |
| | **Average** | 14.0 | 29.7 | 28.3 | **34.2** |
| Action | halfcheetah (2%) | 24.3±1.4 | 13.9±1.9 | 16.2±0.8 | **32.8**±1.2 |
| | hopper (2%) | 36.7±9.6 | **51.9**±3.9 | 37.5±6.0 | 45.3±3.2 |
| | walker2d (2%) | 30.7±7.0 | 27.5±5.3 | 38.9±2.9 | **43.8**±7.9 |
| | **Average** | 30.5 | 31.1 | 30.9 | **40.6** |
| Reward | halfcheetah (2%) | 27.4±2.9 | 36.1±1.3 | 33.3±1.0 | **37.5**±2.0 |
| | hopper (2%) | 32.5±17.5 | **55.5**±6.8 | 51.6±2.7 | 53.9±4.6 |
| | walker2d (2%) | 28.2±18.4 | 42.1±13.4 | 50.5±5.1 | **62.1**±4.7 |
| | **Average** | 29.3 | 44.6 | 45.1 | **51.2** |
| Average over all tasks | | 24.6 | 35.1 | 34.8 | **42.0** |

Table 9: Results on "expert-v2" dataset under random data corruption.

| Attack | Task | DeFog | RIQL | DT | **RDT** |
|---|---|---|---|---|---|
| State | halfcheetah (2%) | 3.8±2.3 | 0.5±1.2 | 2.9±0.8 | **4.4**±0.2 |
| | hopper (2%) | 15.1±4.2 | 32.0±4.4 | 38.5±7.4 | **48.6**±7.0 |
| | walker2d (2%) | 15.3±3.1 | 21.7±4.6 | 40.7±4.8 | **41.6**±4.2 |
| | **Average** | 11.4 | 18.1 | 27.4 | **31.5** |
| Action | halfcheetah (2%) | 2.7±1.0 | -1.3±1.5 | 3.2±2.4 | **5.4**±0.7 |
| | hopper (2%) | 18.0±4.7 | 32.4±8.7 | 21.2±3.3 | **36.0**±4.8 |
| | walker2d (2%) | 25.6±3.8 | 7.1±1.0 | 33.1±6.8 | **73.8**±3.6 |
| | **Average** | 15.5 | 12.7 | 19.2 | 38.4 |
| Reward | halfcheetah (2%) | 2.5±0.8 | 2.7±0.9 | 4.9±0.7 | **17.2**±5.5 |
| | hopper (2%) | 21.2±6.3 | **89.5**±4.6 | 49.5±3.3 | 67.3±6.6 |
| | walker2d (2%) | 36.0±9.8 | 64.2±11.0 | 96.3±2.6 | **103.1**±3.6 |
| | **Average** | 19.9 | 52.1 | 50.2 | **62.5** |
| Average over all tasks | | 15.6 | 27.6 | 32.3 | **44.1** |

## D.12 Performance under NeoRL Dataset

To further demonstrate the robustness of the RDT, we evaluate it on the NeoRL benchmark (Qin et al., 2022). Specifically, we select the "citylearn-medium" and "finance-medium" datasets as the testbeds. The Finance dataset enables the construction of a trading simulator, while the CityLearn dataset reshapes the aggregation curve of electricity demand by controlling energy storage. We randomly sample 20% and 10% of the Finance and CityLearn datasets, respectively, resulting in both datasets having a similar size of approximately $2 \times 10^4$ transitions. We consider RIQL, DeFog, and DT to be comparable baselines. The performance is measured using the response reward criterion, as detailed in Table 10. Notably, RDT achieves the best average performance compared to the other baselines, outperforming DT by 12.13% under Finance dataset. These results underscore RDT's robustness and potential for real-world applications.

Table 10: Results on NeoRL dataset under random data corruption.

| Task | Attack | DeFog | RIQL | DT | **RDT** |
|---|---|---|---|---|---|
| citylearn (20%) | State | **39138.9** | 30235.3 | 37878.9 | 39124.2 |
| | Action | 38970.6 | 29093.2 | 35589.0 | **39066.4** |
| | Reward | 38156.4 | 29634.8 | 37626.9 | **38909.8** |
| Average | | 38755.3 | 29654.4 | 37031.6 | **39033.5** |
| finance (10%) | State | 402.2 | 339.4 | 411.4 | **443.0** |
| | Action | 386.6 | 400.6 | 352.6 | **452.1** |
| | Reward | 330.3 | 376.3 | 329.5 | **417.8** |
| Average | | 373.0 | 372.1 | 347.5 | **437.6** |

### D.13 ADDITIONAL EVALUATION ON ITERATIVE DATA CORRECTION

We have demonstrated the effectiveness of iterative data correction by illustrating the MSE loss between corrected and original data in Figure 3(c). The MSE loss gradually decreases as training proceeds, indicating that RDT can accurately predict the correct data. To further investigate the effectiveness of iterative data correction, we record the precision of detection using the $z$-score. Specifically, **DT w. IDC** can detect $N$ data points as corrupted within a batch, and $M$ out of these $N$ data points are truly corrupted. We record the ratio of $M/N$ under action attack as shown in Figure 18. As observed, the detection precision increases with the training process and can reach 80% to 90% at the 50th epoch, demonstrating the effectiveness of IDC. It is worth noting that the precision is low at the start of training; therefore, we begin data correction at the 50th training epoch in our default implementation.

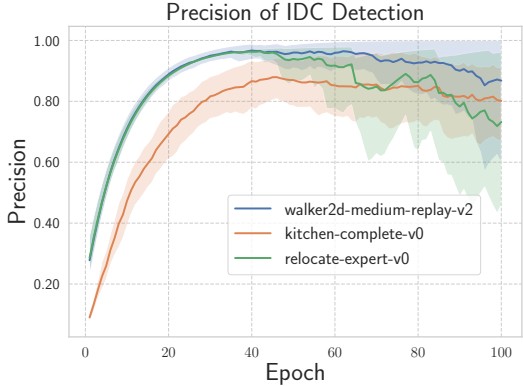

Figure 18: Accuracy of iterative data correction detection via the $z$-score.

### D.14 DISCUSSION ON STATE PREDICTION AND CORRECTION

In the implementation of RDT, we predict both actions and rewards to tackle data corruption. We justify that reward prediction can provide benefits under state and action attacks (see Appendix D.2). Here, we explore the impact of state prediction. We create a variant, **DT w. SP**, based on the original DT, which additionally predicts states. We compare DT and **DT w. SP** under state and action attacks, as shown in Figure 19(a). As observed, state prediction does not consistently benefit all tasks. The performance drops on the Kitchen and Adroit tasks, perhaps due to the larger state dimensions. For example, the state dimension of Kitchen is 60, which is larger than its action dimension of 9.

Based on **DT w. SP**, we evaluate state correction via iterative data correction under different $\zeta$, as shown in Figure 19(b). The results indicate that state correction still causes performance drops. Indeed, state prediction involves learning the environment's dynamics (i.e., transition probabilities), which is inherently challenging, as prediction errors in state prediction negatively impact policy

learning. Additionally, in our experimental setting, data corruption exacerbates prediction errors, leading to further performance degradation. We plan to draw inspiration from model-based RL work to address this issue in future research.

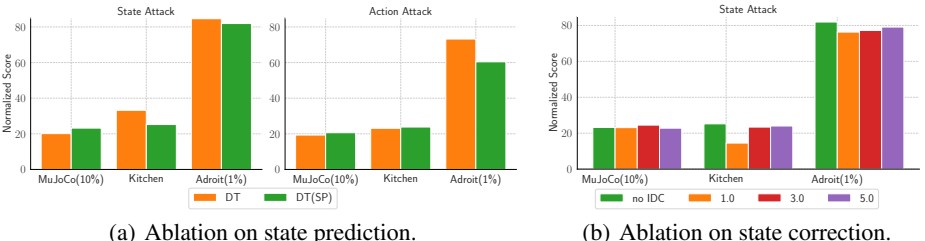

| (a) Ablation on state prediction. | (b) Ablation on state correction. |

Figure 19: Ablation study on state prediction and correction. (a) Comparison of DT and DT with State Prediction (DT w. SP), showing that DT w. SP does not consistently bring improvements. (b) Evaluation of state correction under different $\zeta$ values. Results indicate that state prediction can cause performance drops.

### D.15 MOTIVATING EXAMPLE UNDER DIFFERENT DATA CORRUPTION

To demonstrate the performance drop under varying dataset sizes from another perspective, we modify Figure 1 to create Figure 20. In Figure 20, we place the performances of different algorithms in separate boxes to better illustrate their sensitivity to different types of data corruption. As observed, UWMSG and RIQL are very sensitive to state attacks. This sensitivity arises because we do not separate state and next-state attacks but apply state attacks within a single trajectory, representing a more general setting. UWMSG and RIQL are algorithms based on temporal difference learning, which learns the $Q$-value dependent on the next state. Hence, they experience significant performance drops under state attacks in our settings. In contrast, DeFog and DT, which are based on sequence modeling, are comparatively robust to all types of data corruption. Furthermore, DT demonstrates better robustness than other methods, even with limited datasets. Therefore, we propose three robustness techniques to further improve robustness against data corruption.

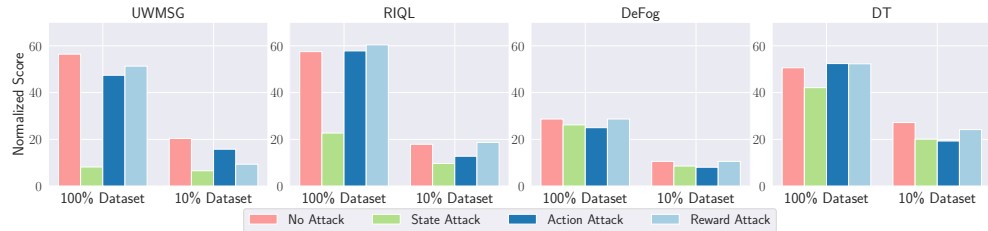

Figure 20: An alternative version of the motivating example, where the performances of different algorithms are placed in separate boxes to better illustrate their sensitivity to various data corruption.

## E LIMITATION AND DISCUSSION

One limitation of RDT is our decision to avoid state prediction and correction within the robust sequence modeling framework. The high dimensionality of states poses significant challenges to achieving accurate predictions. Additionally, data corruption can further complicate state prediction, leading to potential declines in performance, as shown in Appendix D.14. Despite this, we have validated the feasibility and effectiveness of robust sequence modeling in this paper, and we leave state prediction and correction as promising directions for future exploration.

