# OpenReview forum: "Tackling Data Corruption in Offline Reinforcement Learning via Sequence Modeling"
_ICLR.cc/2025/Conference — ICLR 2025 Poster_

### Official Review · Reviewer_GDNb · 2024-10-31

**Soundness:** 2
**Presentation:** 3
**Contribution:** 2
**Rating:** 6
**Confidence:** 4

**Summary:**

This paper introduces a novel method, the Robust Decision Transformer (RDT), designed to address data corruption in offline reinforcement learning. The study finds that traditional offline reinforcement learning methods based on temporal difference learning perform poorly under data corruption, while the DT, a sequence modeling approach, demonstrates greater robustness in such scenarios. RDT enhances DT’s robustness by incorporating three simple yet effective techniques: embedding dropout, Gaussian-weighted learning, and iterative data correction. Experimental results show that RDT significantly outperforms traditional methods across various tasks in the presence of data corruption, especially in complex scenarios where both training and testing phases are affected by attacks.

**Strengths:**

1. Innovation: This approach introduces sequence modeling to address data corruption, moving beyond traditional frameworks based on temporal difference learning.
2. Enhanced Robustness: Embedding dropout, Gaussian-weighted learning, and iterative data correction significantly improve the algorithm’s performance under data corruption.
3. Extensive Validation: The method has been validated across various data corruption scenarios and with limited datasets, demonstrating its robustness.
4. High Adaptability: RDT performs exceptionally well in limited data settings, showcasing the potential of sequence modeling across diverse tasks and data conditions.

**Weaknesses:**

No obvious weaknesses.

**Questions:**

1. Figure 1 is not as intuitive as it could be; it might be more effective to directly display the performance decline under various attacks.
2. In the ablation study, the current approach is to compare RDT with DT(RP) that incorporates three different improvements. A more rational method would be to compare the performance of RDT with RDT w/o (ED, GWL and IDC).

---

> ### Author Response · Authors · 2024-11-20
> **Author Response**
>
> Thank you for the valuable comments, and we provide clarification to your concerns as follows. We hope these new experimental results and explanations can address your concerns, and we look forward to your further feedback.
>
> 1. **‘Figure 1 is not as intuitive as it could be; it might be more effective to directly display the performance decline under various attacks’. (Q1)**
>
>     Thank you for your question. In Figure 1, we aim to highlight the significant performance drop that occurs when using limited data (10%) compared to the full dataset (100%). Therefore, we present the performance of different methods using both the 100% and 10% datasets in a single figure to illustrate more clearly the advantage of DT when dealing with limited data.
>
>     To address the reviewer's concern and further illustrate how performance varies under different data corruption scenarios, we have included another version of Figure 1 in **Appendix D.15 and Figure 20**. These figures depict a more pronounced drop in performance due to state corruption, while the other two types of corruption result in relatively smaller declines, particularly for sequence modeling methods.
>
> 2. **‘A more rational method would be to compare the performance of RDT with RDT w/o (ED, GWL and IDC)’. (Q2)**
>
>     We conduct the ablation study using DT(RP) w. (ED, GWL and IDC) to better demonstrate the impact of the components while mitigating the influence of other factors. Since two components’ interaction can bring unintended influence.
>
>     To address the reviewer's concerns, we conduct additional comparisons with RDT variants that exclude ED, GWL, and IDC. The results of these new ablation studies can be found in **Appendix D.9 and Figure 16**. Notably, these results align with the findings in Section 4.4. This consistency reinforces our initial conclusion that integrating all the proposed techniques (ED, GWL, and IDC) is essential for achieving optimal robustness. The removal of any component leads to a performance drop, with GWL and ED being particularly critical components.

---

> > ### Comment · Reviewer_GDNb · 2024-11-30
> >
> > Thank you to the authors for their response. The additional content provided will greatly enhance the completeness of this paper. As a result, I have increased my confidence.

---

> > > ### Author Response · Authors · 2024-11-30
> > >
> > > We are pleased to see that our rebuttal has addressed the Reviewer's concerns. We appreciate the Reviewer's feedback and their increased confidence in our work.

---

> ### Author Response · Authors · 2024-11-25
>
> We sincerely thank you for your valuable suggestions and time spent on our work. We hope that we have addressed your main questions in the rebuttal. As the due date (November 26 at 11:59pm AoE) for the discussion period is approaching, we hope to receive your further feedback and address your concerns once again. We are committed to addressing any concerns or suggestions you may have to enhance the manuscript.

---

### Official Review · Reviewer_jkcs · 2024-11-02

**Soundness:** 3
**Presentation:** 3
**Contribution:** 3
**Rating:** 8
**Confidence:** 3

**Summary:**

This paper attacks the problem of learning a policy in an offline RL setting, in the specific case of having access to a lossy dataset, where data can be perturbed. While the literature has focused on algorithms leveraging the Q function to handle this problem, this paper shows that sequence-based models such as the Decision Transformer (DT) perform competitively, and are surprisingly robust compared to baselines when the size of the dataset is significantly reduced. The authors then propose 3 modifications to the DT algorithm that lead to algorithm even more robust than DT. They evaluate the performance of their proposed algorithm in the D4RL benchmark, and perform several ablations to get a better understanding of their method.

**Strengths:**

This paper has several strengths.
- I found the introduction did a good job of introducing the problem and motivating the research direction. This is particularly true for the introduction, where the overarching research question was clearly stated. Overall, the paper was well written and pleasant to read. The structure of the document is logical and progressively builds the proposed method. The different contributions were mostly clearly presented, using equations when necessary to support the text. Finally, Fig. 2 was a welcome addition to summarize the modifications brought by the algorithm to the DT architecture.
- Offline RL is an important problem, with many applications in the real-world; I think the community benefits of works improving the performance of policies in that setting, and even more since it includes using a new policy architecture. I find it very sensible to wonder how sequence models fare in that setting, especially since their influence and importance in the literature has grown steadily over the past years. The modules to add to the DT are well motivated and sensible (especially the first 2). In addition, the modules seem rather easy to implement, which makes the paper easily applicable.
- I found the paper did a good job at studying comprehensively the use of DT-like methods in the offline RL literature. Several ablations are done to isolate the contribution of the different elements of RDT (Fig. 3 and Fig. 6). Moreover, there is a large variety of additional experiments in the appendix that naturally answered some of my questions as I read the main text, such as the influence of the value of $\zeta$ and $\beta$ (Fig. 10) or the variation of performance for different corruption scale values (Fig. 11).
- The proposed method delivers, as it gets robust performance in the small scale version of the D4RL dataset, improving over the baselines considered (Table 1), sometimes significantly.
- Ultimately, I think the paper asks an interesting question in the use sequence-based methods for robust offline learning. I believe this can generate interesting discussions at the conference around the usage of sequence-based models.

**Weaknesses:**

There are a couple of axes where I found weaknesses in this work, which I present below.
- Several of the methodological details are pushed to the appendix without being explicitly referred to in the main text. For instance, it is hinted in the main text that the choice of the $\zeta$ parameter (L301) or when the data iterative process starts (L292, “after sufficient training”) is important, but you have to go to the appendix to discover that this information is indeed provided. Adversarial corruption is mentioned quickly (L132) but without giving an intuition about how it would be used in this setting; the action diff perturbation (L426-428) could also have used some intuition (generic is OK) about it before being pushed to the appendix. Overall, I find the authors could make the main text more self-contained by giving the conclusions (maybe broadly) of their different experiments in the main text and referring to the appendix only for the in-depth analysis of the results. Methodological details can be pushed to the appendix, but as the reader goes through the paper, they should know that these details do, in fact, exist.
- While the three proposed additions to DT presented in Sec. 3 make sense, I found their concrete impact on performance difficult to evaluate. For instance, in the ablation of Fig. 6, the addition of the different modules to DT do not drastically change the performance — in fact, in several settings the confidence interval of RDT intersects with the DT(RP). I would be also curious to know how these ablated methods compare with unmodified DT in Fig. 6. Moreover, it seems that different modules contribute more in different settings (ED on action attack, GWL on reward attack …); I think some analysis about which environment should benefit from which module the most, if possible, would have helped framing the modules. The fact that the different ablations of Fig. 6 and 3 are evaluated on different environments also makes it difficult to get a clear sense of the general behavior of each module.
- I find the setting in which this method is expected to perform well to be relatively narrow, as it is mostly valid for datasets that are simultaneously small and corrupted. I do not think this makes this submission less valuable, but it raises questions that I would have liked to see addressed — for instance, what aspect exactly of the low data regime favors sequence models over Q-function based models? Is there any hope for sequence models to overcome Q-function models, even with bigger datasets (especially since transformer models have been shown to work so great in large data regimes)? On this point, while the problem is well-motivated, I found the conclusion that DT worked better than Q-function based models could have used more insights. It is explained at the end of Sec 3.1 (L187-189) but I found these two sentences did not bring much besides repeating the definition of a sequence model.
- Only RDT has an uncertainty measure in Table 1, and there is no uncertainty measure in Table 2. Moreover, I did not find a description of the nature of the uncertainty measure in the main text.

In addition, I list below other minor weaknesses which did not impact my rating, but that I think could be addressed to improve the document:
- The text of the figures is small and makes reading the paper difficult on paper. (Especially true for Fig. 3 that was very hard to read). It would be useful to increase the size of the captions to make the figure more readable.
- L. 98: “Different from the MDP framework, DT [..]”. I would argue that DT still operates within the MDP framework, it is just a different algorithm to learn a policy.
- L246, Eq 4: you did not define the circle-cross operator. If this is supposed to be an elementwise product, I would argue that this is not the most appropriate symbol (\odot would, I think, be more appropriate).
- L535: I found the term “exceptional” to be rather out of place.
- Typo: MoJoCo, while it should be MuJoCo (several places).
- Table 1: Lacking information about the meaning of +- in the RDT column. What exact uncertainty measure did you use?
- In the experiments section, I would refer to the papers that introduced the experiments parameters setting (for example, you set the corruption scale to 1., but unless I am mistaken, we see in the appendix only that this value is set the same as previous works).

**Questions:**

- In Section 3.2.2, you add a reward prediction term that predicts the immediate reward. Unless I am mistaken, you tested your model on dense reward environments, where such a prediction makes sense and might be more aligned with the environment goal. Do you expect your method to work well in non-dense reward environment? If not, do you think “dense reward signal” should be added as a constraint of your method?
- In Section 3.2.2 still, what happens at the beginning of training, when all examples are unknown? How do you differentiate corrupted examples from natural ones, when we expect the model’s prediction to be poor for all inputs?
- Section 3.2.3 : Have you tested how frequently your assumption was true? I am curious to know the proportion of times the z score actually detects the perturbed transitions.
- In Table 1, only RDT gets an uncertainty measure. Why is that?
- In Figure 3, different environments are chosen to showcase the effects of the elements forming the robust DT method (which are also different from the ones of the ablation in Fig. 6). How were these environments chosen? Why not use the same environment? If there is a difference in behavior of the RDT additions across environments, is there a pattern?
- In Figure 6, could you add DT without any modification as a baseline?

---

> ### Author Response · Authors · 2024-11-20
> **Author Response**
>
> Thank you for the valuable comments, and we provide clarification to your concerns as follows. We hope these new experimental results and explanations can address your concerns, and we look forward to your further feedback.
>
> 1. **‘Several of the methodological details are pushed to the appendix without being explicitly referred to in the main text…’. (W1)**
>
>     Thank you for the suggestion. To enhance readability, we have incorporated the recommended modifications for key hyperparameters in the main text of the revised version. For instance, line 303 now includes the choice of $\zeta$ and refers to the implementation of RDT in Appendix C.3. Additionally, we have provided a brief explanation of ‘adversarial corruption’ on line 133 and the ‘action diff’ attack on line 431.
>
> 2. **‘ I would be also curious to know how these ablated methods compare with unmodified DT in Fig. 6…The fact that the different ablations of Fig. 6 and 3 are evaluated on different environments also makes it difficult to get a clear sense of the general behavior of each module’. (W2 and Q6)**
>
>     We have included the comparison results between DT(RP) and the original DT in Appendix D.2, where DT(RP) demonstrates significant improvements in handling state and action corruption. For instance, DT(RP) enhances DT's performance from 84.6 and 73.2 to 93.2 and 92.6, respectively, on state and action corruption in the Adroit tasks. This suggests that the improvement of RDT over DT is more substantial. Since RDT is based on DT(RP), we chose it as the baseline for our ablation study to better understand the impact of the other three techniques.
>
>     We agree that some techniques provide significant improvements under specific data corruption conditions. However, in real-world scenarios, we may not know the specific conditions beforehand. Therefore, our goal is to design an algorithm that is robust against various types of data corruption. From Figure 6, it is evident that the three proposed robustness techniques benefit different tasks. Integrating all these techniques for RDT is essential for achieving strong robustness.
>
>     Regarding Figure 3, we select representative tasks to provide clearer visualization of how the different components function. To address the reviewer's concerns, we rerun the experiments for Figure 3, ensuring the same tasks (walker2d, kitchen-complete, and relocate) presented in Figures 6 and 3. We find the observations remain consistent.
>
> 3. **‘Is there any hope for sequence models to overcome Q-function models, even with bigger datasets’. (W3)**
>
>     In Appendix D.10, we present new experiments conducted with varying dataset sizes, increasing to five and ten times the training data for MuJoCo and Adroit tasks, respectively. Our empirical findings indicate that:
>
>     (1) The overall performance of all methods improves as the dataset size increases.
>
>     (2) Moreover, temporal-difference-based methods like RIQL are comparable to vanilla sequence modeling methods like DT when the dataset is large, but they are less preferable when the dataset size is limited.
>
>     (3) Notably, RDT consistently outperforms other baselines across all dataset sizes, validating the effectiveness and importance of our proposed robustness techniques.
>
>     For your reference, we provide the results for the Adroit task below.
>
>     #### **Results under random data corruption on the Adroit tasks with various dataset scales.**
>     | Attack | Task               | DeFog            | RIQL            | DT              | **RDT**          |
>     |--------|--------------------|------------------|-----------------|-----------------|------------------|
>     |State   | Adroit(1%)  |69.7 |38.1 |64.9 |93.9 |
>     |        | Adroit(5%)  |77.5 |72.9 |97.3 |99.8 |
>     |        | Adroit(10%) |85.1 |80.2 |98.0 |106.7|
>     |Action  | Adroit(1%)  |81.8 |39.6 |73.2 |96.5 |
>     |        | Adroit(5%)  |86.5 |85.4 |76.6 |103.7|
>     |        | Adroit(10%) |94.4 |103.6|101.9|109.7|
>     |Reward  | Adroit(1%)  |85.7 |39.1 |83.8 |104.1 |
>     |        | Adroit(5%)  |89.5 |94.2 |92.6 |104.9 |
>     |        | Adroit(10%) |94.6 |101.0|96.5 |108.1 |
>
>
> 4. **‘Only RDT has an uncertainty measure in Table 1, and there is no uncertainty measure in Table 2. Moreover, I did not find a description of the nature of the uncertainty measure in the main text’. (W4, Q4)**
>
>     We repeat all the experiments using four different random seeds and calculate the standard deviation as a measure of uncertainty. While we choose not to include the uncertainty of baselines in Table 1 to maintain simplicity and readability, we omitted the variance measure in Table 2 because it averages over each task group (MuJoCo, Kitchen, and Adroit), making the variance less meaningful. **Readers can refer to Tables 5 and 6 in Appendix D.6 for the raw results with reported variance.**

---

> ### Author Response · Authors · 2024-11-20
> **Author Response**
>
> 5. **‘Do you expect your method to work well in non-dense reward environment’. (Q1)**
>
>     In our experiments, the Adroit and Kitchen tasks are both examples of sparse reward tasks. Interestingly, as shown in Appendix D.2, DT(PR) consistently improves performance under state and action corruption for these tasks. Although DT(PR) may decrease performance under reward corruption, other components of RDT, such as GWL, ED, and IDC, can enhance performance in these scenarios, which are not dependent on dense reward signals. Therefore, a "dense reward signal" is not a necessary requirement for our RDT.
>
> 6. **‘In Section 3.2.2 still, what happens at the beginning of training, when all examples are unknown? How do you differentiate corrupted examples from natural ones, when we expect the model’s prediction to be poor for all inputs?’. (Q2)**
>
>     As shown in Figure 3(b), we plot the loss for clean and corrupted data separately for DT and DT w. GWL. We find that the model quickly differentiates between clean and corrupted data, with an evident gap between the two losses from the very first evaluation point. This disparity suggests that applying GWL effectively prevents the model from overfitting to corrupted data without negatively impacting the learning from clean data. Additionally, for Iterative Data Correction, we initiate the process midway through the training to avoid potential non-differentiable cases at the start of training.
>
> 7. **‘Section 3.2.3 : Have you tested how frequently your assumption was true? I am curious to know the proportion of times the z score actually detects the perturbed transitions’. (Q3)**
>
>     To further investigate the effectiveness of IDC, we record the accuracy of detection via the $z$-score in Appendix D.13 of our revision. Specifically, \textbf{DT w. IDC} can detect $N$ data points as corrupted within a batch, and $M$ out of these $N$ data points are truly corrupted. We record the ratio of $M/N$ under action attack as shown in Figure 18. As observed, the detection accuracy increases with the training process and reaches nearly 80% ~ 90% at the 50th epoch, demonstrating the effectiveness of IDC. It is worth noting that the accuracy is relatively low at the start of training; therefore, we begin Iterative Data Correction at the middle of training in our default implementation.
>
> 8. **‘In Figure 3, different environments are chosen to showcase the effects of the elements forming the robust DT method (which are also different from the ones of the ablation in Fig. 6). How were these environments chosen? Why not use the same environment? If there is a difference in behavior of the RDT additions across environments, is there a pattern?’ (Q5)**
>
>     In Figure 3, we select representative tasks to provide clearer visualization of how the different components function. To address the reviewer's concerns, we rerun the experiments for Figure 3, ensuring that all three subfigures feature the same tasks (walker2d, kitchen-complete, and relocate) and are consistent with those presented in Figure 6. We find that the observations remain consistent.
>
> 9. **Minor weaknesses.**
>
>     Thank you for pointing out these minor weaknesses in our manuscript. We have made corresponding modifications to address them in our revised manuscript.

---

> > ### Comment · Reviewer_jkcs · 2024-11-22
> >
> > 5.
> > I see, thanks for correcting me about the existence of the sparse reward tasks.
> > It is true that Fig. 9 does show a consistent improvement of DT(RP) compared to DT on the kitchen and adroit tasks.
> > I am not sure to understand why the reward prediction helps. Could you please explain to me why this is the case, given that there is very little signal to learn from in these settings?
> > Moreover, while the improvement is consistent (which leads me to indeed believe the better performance of DT(RP)), the plot is lacking a measure of uncertainty of the estimates. Did you run several seeds to confirm further these results?
> >
> > 6.
> > Thanks for the answer. I don’t have any further question on this topic at this time.
> >
> > 7.
> > Thank you for running this experiment! It is interesting that the detection works so well.
> > Just a correction: I believe you are not measuring the accuracy but the precision (true positive / (all predicted positives)) of your method. I would suggest to correct the denomination.
> >
> > 8.
> > Thank you for making this additional experiment and checking that the results were consistent.
> >
> > 9.
> > Thanks!

---

> > > ### Author Response · Authors · 2024-11-24
> > > **Author Response**
> > >
> > > Thank you for your response and for acknowledging that we have addressed some of your concerns. We are pleased to address your remaining concerns below.
> > >
> > > 2. **‘I still think that DT should be the baseline in Fig. 6 rather than DT(RP), given that it is your actual baseline and we would like to see how all modifications compare. For instance, the performance of DT(RP) degrades compared to DT on the reward attack… Could you please include DT in Fig. 6, including uncertainty measures?’**
> > >
> > >     As the reviewer suggested, we now include the original DT in Figure 6 with an uncertainty measure. For most tasks, DT(RP) shows improvements over or is comparable to the original DT. However, in the Kitchen and Adroit task groups with reward corruption, there is an overall performance degradation. We have also provided the raw data below for your review concerning the reward corruption.
> > >
> > >     Although DT(RP) reduces performance by nearly 50% in the reward attack of Figure 6, integrating DT(RP) with Gaussian Weighted Learning can outperform the original DT, and combining all three components (i.e., RDT) yields even better results.
> > >
> > >     #### **The detailed comparison results under reward corruption**
> > >     | Task                | DT   |  DT(RP) |
> > >     |---------------------|------|---------|
> > >     | halfcheetah (10%)| 9.3±0.9 | 9.5±0.9 |
> > >     | hopper (10%)     | 36.0±7 | 40.5±4.3 |
> > >     | walker2d (10%)   | 27.4±3 | 28.6±4.3 |
> > >     | **Average** | 24.2 | 26.2 |
> > >     | kitchen-complete | 43.9±4.3 | 52.4±3.1 |
> > >     | kitchen-partial  | 47.1±6.9 | 0.1±0.2 |
> > >     | kitchen-mixed    | 52.2±1.9 | 0.1±0.2 |
> > >     | **Average** | 47.8 | 17.5 |
> > >     | door (1%)     | 99.0±2.3| 101.2±1.8  |
> > >     | hammer (1%)   | 80.7±11.2 | 102.5±4.7  |
> > >     | relocate (1%) | 71.8±7.7 | 23.7±7.5   |
> > >     | **Average** | 83.8 | 75.8 |
> > >
> > >
> > > 3. **‘I wonder how RDT would fare in the 100% dataset regime, which is the one that was used in Fig. 1 to motivate the focus on small datasets. In particular, are the modifications that RDT brinm sufficient to beat Q methods in all settings, even when the dataset is *not* limited? …  whether they expect this improved performance of RDT to be sustained to regular sized datasets like the ones usually found in the offline RL literature**.**’**
> > >
> > >     Thank you for the question. Our findings (in the below table) indicate that RDT can outperform or  match the performance of temporal difference based methods, such as RIQL, when using a 100% dataset, with particular advantages under state corruption .
> > >
> > >     An intuitive explanation for the impact of dataset size is that larger datasets can often contain multiple instances of a transition, thereby increasing the probability that not all instances are corrupted. This allows robust methods like RIQL to learn easily from the high-confidence instances. Conversely, in smaller datasets, a transition might appear only once, and if it is corrupted, it becomes more challenging for offline RL methods to learn a correct policy. As a result, all methods tend to experience a significant performance drop when data is limited, which is why we primarily focus on the more challenging, limited data setting in our study.
> > >
> > >     #### **Results under random data corruption on the MuJoCo tasks with various dataset scales.**
> > >
> > >     | Attack | Task                | RIQL            | DT              | **RDT**          |
> > >     |--------|--------------------|------------------|-----------------|-----------------|
> > >     |State   | MuJoCo(10%) |  9.7 | 16.8| **23.1**|
> > >     |        | MuJoCo(20%) | 17.9| 24.4| **29.3**|
> > >     |        | MuJoCo(50%) | 27.4| 32.5| **38.3**|
> > >     |        | MuJoCo(100%)|  22.7| 42.1| **46.9**|
> > >     |Action  | MuJoCo(10%) | 12.8| 19.3| **29.5**|
> > >     |        | MuJoCo(20%) | 32.3| 26.3| **37.2**|
> > >     |        | MuJoCo(50%) | 42.3| 44.2| **50.4**|
> > >     |        | MuJoCo(100%)| **57.8**| 52.4| 54.0|
> > >     |Reward  | MuJoCo(10%) |18.6| 24.2| **31.5**|
> > >     |        | MuJoCo(20%) |  32.8| 31.6| **40.1**|
> > >     |        | MuJoCo(50%) |  50.8| 46.0| **53.6**|
> > >     |        | MuJoCo(100%)|  **60.4**| 52.3| 59.0|

---

> > > ### Author Response · Authors · 2024-11-24
> > > **Author Response**
> > >
> > > 4. **‘I do not understand why you cannot also compute the uncertainty over this average. … I kindly encourage you to include the full results, including the uncertainty estimates, in the main part of the document.’**
> > >
> > >     Thanks for the suggestion. We now include variance for all algorithms in Table 1. We would like to clarify why results (average over task group) like Table 2 do not include an uncertainty measure, whereas results like Table 1 do. The **uncertainty measure is meaningful when evaluating multiple runs with different random seeds on a single task**. However, when considering the performance across a group of tasks—such as those in the MuJoCo task group, which includes three distinct tasks—**an uncertainty measure across the entire group can be meaningless because of the inherent differences in rewards among different tasks**.
> > >
> > >     Averaging over task group is commonly adopted in the literature **to simplify data presentation** when dealing with diverse task groups, as seen in Figure 3 of the DT paper [1] and Figure 1 in the Lamo paper [2]. Due to the limited space in the main paper, we have provided the raw data with variance for Table 2 in Appendix D.6, which contains results for each individual task within the task group.
> > >
> > >
> > >
> > > 5. **‘I am not sure to understand why the reward prediction helps. Could you please explain to me why this is the case, given that there is very little signal to learn from in these settings? Moreover, while the improvement is consistent (which leads me to indeed believe the better performance of DT(RP)), the plot is lacking a measure of uncertainty of the estimates. Did you run several seeds to confirm further these results?’**
> > >
> > >     Intuitively, reward prediction serves as an auxiliary task to prevent the model from overfitting to erroneous context-to-action mapping. For example, when state or action is corrupted but the reward is correct, this prediction acts as a form of regularization, limiting excessive shifts in policy towards incorrect labels. Conversely, if the reward is corrupted, it may negatively affect action prediction, necessitating the use of other robust techniques in RDT.
> > >
> > >     The sparse reward task, Kitchen, has a special design, where, after each subgoal is achieved, all subsequent rewards are incremented by 1, with a maximum of four subgoals. As a result, the rewards in this dataset range from [0, 4]. The average rewards in the two Kitchen datasets are 1.35 and 1.68, respectively. Thus, while Kitchen has sparse rewards in comparison to other tasks in D4RL, they are not excessively sparse. RDT can leverage this relatively sparse reward to enhance performance under data corruption.
> > >
> > >     As detailed in Section 4.1 and Appendix C.3, all experiments in our study were conducted with 4 different random seeds. We have also explained why we do not report uncertainty over the task group in the above response. The average results for the task group provide readers with a clearer understanding of the overall findings, which is particularly beneficial for the ablation study. However, if the reviewer thinks it necessary, we can provide a table of raw results for the ablation study.
> > >
> > > 7. **‘I believe you are not measuring the accuracy but the precision (true positive / (all predicted positives)) of your method. I would suggest to correct the denomination.’**
> > >
> > >     Thank you for pointing out this issue. We have modified the description in Appendix D.13 accordingly.
> > >
> > > [1] Chen L, Lu K, Rajeswaran A, et al. Decision transformer: Reinforcement learning via sequence modeling. Advances in neural information processing systems, 2021.
> > >
> > > [2] Shi R, Liu Y, Ze Y, et al. Unleashing the power of pre-trained language models for offline reinforcement learning. ICLR, 2024.

---

> > > > ### Comment · Reviewer_jkcs · 2024-11-25
> > > >
> > > > 2.
> > > > Thank you for adding DT to Fig. 6, this is the Figure I was hoping to see.
> > > > I do find that for individual tasks, the performance improvements of RDT and the ablations are in several instances rather small compared to the baseline DT (the prediction intervals intersect).
> > > > However, the results do show a performance increase of RDT compared to its ablations and the baseline, which is consistent across tasks.
> > > > I have no further questions on this point.
> > > >
> > > > 3.
> > > > Thank you for improving further the experimental part of the paper by including the 100% results.
> > > > The discussion makes sense to me! I have no further question.
> > > >
> > > > 4.
> > > > Thank you for your answer.
> > > > I do not agree with the reasoning: in my opinion, if you deem that the reward differences across tasks prevent you from getting a reasonable uncertainty estimate, then you should not be comparing them to begin with. If they are comparable enough to have a meaningful statistic compared over them, I see no reason why you cannot compute a equally meaningful uncertainty interval over this statistic.
> > > >
> > > > That said, as you mention, you do provide the full per-task results with their variance in the appendix.
> > > > Therefore, I do not believe it is constructive for me to insist further on this rather small point, as you provide the information in some form.
> > > >
> > > > 5.
> > > > I see. In that case, I believe it could be interesting for future work to consider what is the robustness of RDT to tasks with a really sparse reward signal — though I assume this is an information you could have before deployment.
> > > > I think the sentence you used higher up to summarize is fair: “Therefore, a "dense reward signal" is not a necessary requirement for our RDT.”, since a partially dense reward signal seems to be OK too.
> > > > I have no further question on this point.
> > > >
> > > >
> > > > Thanks again for the improvements brought to the paper during the rebuttal. I have raised my score in consequence.

---

> > > > > ### Author Response · Authors · 2024-11-26
> > > > >
> > > > > We are pleased to observe that our rebuttal has satisfactorily addressed the Reviewer's concerns. We would like to express our gratitude to the Reviewer for both their constructive feedback and for raising their score. In our future revisions, we will include the discussion on reward prediction and other recommended content.

---

> ### Comment · Reviewer_jkcs · 2024-11-22
>
> I want to thank the authors for the rebuttal, the modifications already done and the added experiments. Please find below my subsequent comments, with a few additional questions:
>
> 1.
> Thank you for doing the modifications.
>
> 2.
> Thanks for the additional explanations. I still think that DT should be the baseline in Fig. 6 rather than DT(RP), given that it is your actual baseline and we would like to see how all modifications compare. For instance, the performance of DT(RP) degrades compared to DT on the reward attack. You explain that the 3 modules improve on it, so I would like to see all the results on the same Figure to be able to appreciate this exactly.
> In addition, Fig. in Appendix D2 does not include uncertainty measures while Fig. 6 does.
> Could you please include DT in Fig. 6, including uncertainty measures?
>
> 3.
> Thank you for running this additional experiment! It is a very positive sign that the performance of RDT consistently improves over the other methods.
> I wonder how RDT would fare in the 100% dataset regime, which is the one that was used in Fig. 1 to motivate the focus on small datasets. In particular, are the modifications that RDT brinm  sufficient to beat Q methods in all settings, even when the dataset is *not* limited?
> To be clear, I do not expect the authors to run additional experiments (though additional results are always welcome!), but rather to discuss whether they expect this improved performance of RDT to be sustained to regular sized datasets like the ones usually found in the offline RL literature.
>
>
> 4.
> Sorry, but I am not sure I agree with the reasoning.
> If you deemed these quantities sufficiently similar to average them together in this table, I do not understand why you cannot also compute the uncertainty over this average.
> Moreover, I do not agree that the information about the uncertainty of the performance estimates is superfluous in the main text. I kindly encourage you to include the full results, including the uncertainty estimates, in the main part of the document.

---

### Official Review · Reviewer_bfdB · 2024-11-04

**Soundness:** 2
**Presentation:** 3
**Contribution:** 3
**Rating:** 6
**Confidence:** 4

**Summary:**

This paper focus on invesgating the robustness of offline RL to various data corruption, including states, actions, rewards.  The problem getting more challengable as the dataset is limited. To tackle this problem, the paper proposed Robust Decision Transformer (RDT) by incorporating three effective components, embedding dropout, gaussian weighted learning and iterative data correction. The experiment results demonstrates RDT consistently outperforms the baselines on most tasks under limited corrupt dataset settings.

**Strengths:**

1. This paper is well-written, and the main idea is easy to follow.
2. The problem setting is interesting, and the proposed method is straightforward, effectively demonstrating its efficiency in handling corrupted, limited datasets.

**Weaknesses:**

1. There is no discussion regarding corrupted data at different scales. An analysis of how dataset scale affects the challenges of data corruption would be beneficial.
2. The proposed method is only evaluated on the medium-replay dataset for MuJoCo tasks and the expert dataset for Adroit. A discussion on the impact of corrupted data with varying quality levels is needed.
3. No discussion of the limitations of RDT, which would help in fairly assessing the proposed approach
4. There is a typographical error on line 452: "relocate-exper" should be corrected to "relocate-expert."

**Questions:**

1. In the gaussian weight learning section, the paper proposes using the exponential form of the prediction loss as a weight for the optimization terms in RDT. This design seems questionable, as early in training, the model is likely to be inaccurate, resulting in high loss values. However, this approach down-weights these losses, potentially reducing gradient information and slowing the learning process. Could the authors provide more intuition behind the design of these weighted terms?
2. Based on the experimental results, RDT appears to provide greater benefits for Adroit tasks compared to the other two task settings. Could you explain the reason for this difference?
3. I reviewed the full dataset results in RIQL, and they seem to conflict with the scores listed in the paper’s appendix. Could you clarify why this discrepancy occurs?

---

> ### Author Response · Authors · 2024-11-20
> **Author Response**
>
> Thank you for the valuable comments, and we provide clarification on your concerns as follows. We hope these new experimental results and explanations can address your concerns, and we look forward to your further feedback.
>
> 1. **‘An analysis of how dataset scale affects the challenges of data corruption would be beneficial’. (W1)**
>
>     Thank you for the insightful question. To provide an intuitive explanation: with larger datasets, a given transition is likely present in the dataset multiple times, increasing the probability that not all instances are corrupted. This makes it easier to learn from clean data. In contrast, in smaller datasets, a transition may exist only once, and if corrupted, it poses a greater challenge for offline RL methods to learn a correct policy. Our results in Figure 1 support this intuition, showing significant performance drops for all methods when limited to 10% of the datasets. Therefore, we primarily focus on the more challenging, limited data setting.
>
>     To address the reviewer’s concern,  we conduct new experiments with various corrupted dataset sizes, detailed in Appendix D.10, to better understand the impact of dataset size on the performance of RDT and other methods. Our empirical findings reveal the following:
>
>     1. The overall performance of all methods improves as the dataset size increases.
>     2. Temporal-difference-based methods like RIQL are comparable to vanilla sequence modeling methods like DT when dataset is large, but are less preferable when the dataset size is limited.
>     3. RDT consistently outperforms other baselines across all dataset sizes, validating the effectiveness and importance of our proposed robustness techniques.
>
>     To provide a convenient reference, we format the following tables extracted from Figure 17.
> #### **Results under random data corruption on the MuJoCo tasks with various dataset scales.**
> | Attack | Task               | DeFog            | RIQL            | DT              | **RDT**          |
> |--------|--------------------|------------------|-----------------|-----------------|------------------|
> |State   | MuJoCo(10%) | 12.9 | 9.7 | 16.8| 23.1|
> |        | MuJoCo(20%) | 13.6 | 17.9| 24.4| 29.3|
> |        | MuJoCo(50%) | 18.6 | 27.4| 32.5| 38.3|
> |Action  | MuJoCo(10%) | 19.0 | 12.8| 19.3| 29.5|
> |        | MuJoCo(20%) | 32.6 | 32.3| 26.3| 37.2|
> |        | MuJoCo(50%) | 46.0 | 42.3| 44.2| 50.4|
> |Reward  | MuJoCo(10%) | 17.1 | 18.6| 24.2| 31.5|
> |        | MuJoCo(20%) | 32.8 | 32.8| 31.6| 40.1|
> |        | MuJoCo(50%) | 41.3 | 50.8| 46.0| 53.6|
> #### **Results under random data corruption on the Adroit tasks with various dataset scales.**
> | Attack | Task               | DeFog            | RIQL            | DT              | **RDT**          |
> |--------|--------------------|------------------|-----------------|-----------------|------------------|
> |State   | Adroit(1%)  |69.7 |38.1 |64.9 |93.9 |
> |        | Adroit(5%)  |77.5 |72.9 |97.3 |99.8 |
> |        | Adroit(10%) |85.1 |80.2 |98.0 |106.7|
> |Action  | Adroit(1%)  |81.8 |39.6 |73.2 |96.5 |
> |        | Adroit(5%)  |86.5 |85.4 |76.6 |103.7|
> |        | Adroit(10%) |94.4 |103.6|101.9|109.7|
> |Reward  | Adroit(1%)  |85.7 |39.1 |83.8 |104.1 |
> |        | Adroit(5%)  |89.5 |94.2 |92.6 |104.9 |
> |        | Adroit(10%) |94.6 |101.0|96.5 |108.1 |

---

> ### Author Response · Authors · 2024-11-20
> **Author Response**
>
> 2. **‘A discussion on the impact of corrupted data with varying quality levels is needed’**. **(W2)**
>
>     We agree that data quality is a critical factor in our benchmark design, and we already consider it in our evaluation. Since we investigate data corruption across three data components—state, action, and reward, and two types of corruption (random and adversarial), the experimental workload is substantial. Therefore, we select one representative dataset for each MuJoCo and Adroit task and all datasets from Kitchen. The *medium-replay* dataset offers a wide range of data quality, from random to medium, reflecting conditions close to real-world scenarios. In a word, **our experiments already cover diverse data quality for comprehensive analysis, including random to medium-quality data in MuJoCo, varying quality of human demonstrations in Kitchen, and expert data in Adroit.**
>
>     To further evaluate the robustness across different data quality, we conduct an additional evaluation using *expert* and *medium* datasets from MuJoCo tasks during the rebuttal. We randomly sample a similar size of approximately $2 \times 10^4$ transitions to match the dataset size of our main experiments. These results are detailed in Appendix D.11 of our revision. Notably, **RDT consistently outperforms other baselines across different quality level datasets**. This superior performance reinforces RDT's capability to handle a range of dataset qualities efficiently.
>
>     We provide the following tables from Appendix D.11 for your convenient reference.
>
>
> #### **Results on "medium-v2" dataset under random data corruption.**
>
> | Attack | Task               | DeFog           | RIQL            | DT              | **RDT**          |
> |--------|--------------------|-----------------|-----------------|-----------------|------------------|
> | State  | halfcheetah (2%)   | 7.5 ± 1.4       | 18.0 ± 1.5      | 15.7 ± 1.3      | **22.3 ± 1.1**      |
> |        | hopper (2%)        | 20.4 ± 8.3      | 45.7 ± 7.3      | 48.6 ± 3.2      | **52.2 ± 5.7**    |
> |        | walker2d (2%)      | 14.2 ± 1.6      | 25.4 ± 5.0      | 20.5 ± 6.0      | **28.0 ± 8.0**      |
> | **Average** |                | 14.0            | 29.7            | 28.3            | **34.2**            |
> | Action | halfcheetah (2%)   | 24.3 ± 1.4      | 13.9 ± 1.9      | 16.2 ± 0.8      | **32.8 ± 1.2**    |
> |        | hopper (2%)        | 36.7 ± 9.6      | **51.9 ± 3.9**    | 37.5 ± 6.0      | 45.3 ± 3.2      |
> |        | walker2d (2%)      | 30.7 ± 7.0      | 27.5 ± 5.3      | 38.9 ± 2.9      | **43.8 ± 7.9**    |
> | **Average** |                | 30.5            | 31.1            | 30.9            | **40.6**            |
> | Reward | halfcheetah (2%)   | 27.4 ± 2.9      | 36.1 ± 1.3      | 33.3 ± 1.0      | **37.5 ± 2.0**    |
> |        | hopper (2%)        | 32.5 ± 17.5     | **55.5 ± 6.8**    | 51.6 ± 2.7      | 53.9 ± 4.6      |
> |        | walker2d (2%)      | 28.2 ± 18.4     | 42.1 ± 13.4     | 50.5 ± 5.1      | **62.1 ± 4.7**    |
> | **Average** |                | 29.3            | 44.6            | 45.1            | **51.2**            |
> | **Average over all tasks** |                 | 24.6            | 35.1            | 34.8            | **42.0**            |
>
>
> #### **Results on "expert-v2" dataset under random data corruption.**
>
> | Attack | Task               | DeFog           | RIQL            | DT              | **RDT**          |
> |--------|--------------------|-----------------|-----------------|-----------------|------------------|
> | State  | halfcheetah (2%)   | 3.8 ± 2.3       | 0.5 ± 1.2       | 2.9 ± 0.8       | **4.4 ± 0.2**    |
> |        | hopper (2%)        | 15.1 ± 4.2      | 32.0 ± 4.4      | 38.5 ± 7.4      | **48.6 ± 7.0**   |
> |        | walker2d (2%)      | 15.3 ± 3.1      | 21.7 ± 4.6      | 40.7 ± 4.8      | **41.6 ± 4.2**   |
> | **Average** |                | 11.4            | 18.1            | 27.4            | **31.5**         |
> | Action | halfcheetah (2%)   | 2.7 ± 1.0       | -1.3 ± 1.5      | 3.2 ± 2.4       | **5.4 ± 0.7**    |
> |        | hopper (2%)        | 18.0 ± 4.7      | 32.4 ± 8.7      | 21.2 ± 3.3      | **36.0 ± 4.8**   |
> |        | walker2d (2%)      | 25.6 ± 3.8      | 7.1 ± 1.0       | 33.1 ± 6.8      | **73.8 ± 3.6**   |
> | **Average** |                | 15.5            | 12.7            | 19.2            | **38.4**         |
> | Reward | halfcheetah (2%)   | 2.5 ± 0.8       | 2.7 ± 0.9       | 4.9 ± 0.7       | **17.2 ± 5.5**   |
> |        | hopper (2%)        | 21.2 ± 6.3      | **89.5 ± 4.6**    | 49.5 ± 3.3      | 67.3 ± 6.6       |
> |        | walker2d (2%)      | 36.0 ± 9.8      | 64.2 ± 11.0     | 96.3 ± 2.6      | **103.1 ± 3.6**  |
> | **Average** |                | 19.9            | 52.1            | 50.2            | **62.5**         |
> | **Average over all tasks** |                 | 15.6            | 27.6            | 32.3            | **44.1**         |

---

> ### Author Response · Authors · 2024-11-20
> **Author Response**
>
> 3. **‘No discussion of the limitations of RDT’. (W3)**
>
>     One limitation of RDT is our decision to avoid state prediction and correction within the robust sequence modeling framework. The high dimensionality of states poses significant challenges to achieving accurate predictions. Additionally, data corruption can further complicate state prediction, leading to potential declines in performance. Despite this, we have validated the feasibility and effectiveness of robust sequence modeling in this paper, and we leave state prediction and correction as promising directions for future exploration. We add the limitation discussion to Appendix E.
>
> 4. **‘as early in training, the model is likely to be inaccurate, resulting in high loss values…provide more intuition behind the design of these weighted terms’. (Q1)**
>
>     Thank you for the insightful question. In our study on learning under data corruption, we find that the most crucial factor affecting performance is not the absolute scale of the loss but the relative difference between losses for clean and corrupted data. In Figure 3(b), we present results for DT and DT w/ GWL, showing the loss for clean data and corrupted data. Notably, even at the first evaluation point during training, there is a clear disparity between the losses associated with clean and corrupted data. This disparity indicates that applying GWL is effective in preventing the model from overfitting to corrupted data while not adversely affecting learning from clean data.
>
>     We think that a better approach might involve setting the temperature hyperparameter $\beta$ dynamically, starting from 0 and increasing to a maximum value during training. This could potentially address the issue highlighted by the reviewer. However, this would add complexity to the method. Therefore, we choose not to include it by default, but it remains a natural and promising extension of RDT.
>
> 5. **‘RDT appears to provide greater benefits for Adroit tasks compared to the other two task settings. Could you explain the reason for this difference’. (Q2)**
>
>     While RDT achieves the highest absolute score on Adroit tasks, it is important to note that RDT demonstrates relative improvement across all three task settings. Specifically:
>
>     - Under random attack conditions, RDT outperforms DT by:
>
>         **32.15%** on MuJoCo, **40.49%** on Kitchen, **21.86%** on Adroit.
>
>     - Under adversarial attack conditions, RDT exceeds DT by:
>
>         **27.44%** on MuJoCo, **52.19%** on Kitchen, **21.74%** on Adroit.
>
>
>     Thus, Adroit does not show the greatest improvement over DT. Our ablation study in Section 4.4  reveals that each component contributes to the enhancement of RDT, with Gaussian Weighted Learning being the most significant, particularly for the Kitchen task. A possible explanation is that long-horizon tasks with sparse rewards, such as Kitchen, require accurate action labels; otherwise, achieving rewards becomes challenging. This is where RDT proves to be more advantageous.
>
> 6. **‘the full dataset results in RIQL, and they seem to conflict with the scores listed in the paper’s appendix’. (Q3)**
>
>     Thanks for the question. We would like to highlight a difference in settings between the original RIQL and ours, as emphasized in line 118. In our unified data corruption, only three independent elements—states, actions, and rewards—are considered under a trajectory-based storage framework. Therefore, our definition of state corruption includes both state and dynamic corruption as defined in RIQL. The original RIQL implementation requires separate parameter tuning for state and dynamics corruption, which is not feasible in our context, potentially leading to a performance drop.
>
>     Additionally, we present an analysis of the RIQL learning curve in Appendix D.7. When comparing this curve to that in the RIQL paper, we find that RIQL requires more epochs (3,000 epochs) to converge in the full dataset setting, while it converges faster (500 epochs) with limited data (10% MuJoCo). Given our focus on the limited data setting, we observe that 1,000 epochs are sufficient for both RDT and baselines like RIQL to converge, as detailed in Appendix D.5. Therefore, we use 1,000 training epochs by default. In the revision, we will include results for the full dataset using three times the training epochs for both RDT and RIQL.
>
> 7. **Typos**
>
>     Thank you for pointing out those typos and minor issues. We have fixed them in the revised version.

---

> > ### Comment · Reviewer_bfdB · 2024-11-21
> > **Thanks for your responses.**
> >
> > I appreciate the authors for providing a more detailed evaluation to demonstrate the effectiveness of RDT in the limited corrupted data setting, along with a thorough explanation that resolves most of my concerns.
> >
> > I have raised my score accordingly. However, I still believe it is unfair to compare RIQL based on results where the algorithm has not fully converged. In offline RL settings, training efficiency is generally less critical than in online RL, making this comparison less representative.
> >
> > Additionally, I have one question for discussion purposes (not requiring further experiments): Have you considered evaluating diffusion-based methods in the corrupted data setting? Given their inherent capacity for "denoising," diffusion models might be well-suited for handling corrupted data.

---

> > > ### Author Response · Authors · 2024-11-24
> > > **Author Response**
> > >
> > > Thank you for raising the score. We appreciate your valuable feedback, which helped improve our manuscript. We are happy to address your remaining concerns and discuss potential future work.
> > >
> > > 1. **‘I still believe it is unfair to compare RIQL based on results where the algorithm has not fully converged’**
> > >
> > >     In Figure 12, we demonstrate that 1,000 epochs are sufficient for RIQL to converge in the limited (10%) data setting, ensuring fair comparisons in our main experiments. To address the reviewer's concerns about the full (100%) data setting on MuJoCo, we rerun RIQL experiments for 3,000 epochs and RDT experiments for 300 epochs, which are 3 times the default.
> > >
> > >     The results show that RIQL benefits significantly from extending to 3,000 epochs, whereas RDT exhibits only marginal improvement. Upon convergence, RIQL performs well under action and reward corruption but underperforms compared to RDT under state corruption. Overall, RDT and RIQL are comparable in the full dataset setting. These findings suggest that **temporal difference-based methods like RIQL are more advantageous with full datasets**, while **robust sequence modeling methods like RDT are stable across varying data sizes but particularly favorable with limited data or state corruption**. We will update Table 6 in Appendix D.7 with the extended training results for other methods in our revised paper.
> > >
> > >
> > >     | Attack | Task                | RIQL (1000 epochs) |RIQL (3000 epochs)|RDT (100 epochs) | RDT (300 epochs)  |
> > >     |--------|---------------------|-------------------|-----------------|----------------|------------------|
> > >     | State  | halfcheetah         | 19.9±2.1   | 28.0±3.0  | 30.8±1.8    | 30.0±2.7    |
> > >     |        | hopper              | 34.0±13.4  | 41.2±24.5 | 56.6±12.9   | 56.6±9.4    |
> > >     |        | walker2d            | 14.2±1.2   | 22.6±8.7  | 53.4±4.0    | 53.3±2.5    |
> > >     | **Average** |                | 22.7       | 30.6      | 46.9        | **47.4**       |
> > >     | Action | halfcheetah         | 41.3±1.4   | 43.5±0.7  | 37.4±1.4    | 37.7±2.2    |
> > >     |        | hopper              | 59.6±12.7  | 80.9±4.8  | 63.2±5.8    | 75.3±7.0    |
> > >     |        | walker2d            | 72.6±17.3  | 87.0±2.5  | 61.4±1.7    | 61.1±3.0    |
> > >     | **Average** |                | 57.8       | **70.5**      | 54.0        | 58.0       |
> > >     | Reward | halfcheetah         | 42.7±0.9   | 43.3±0.9  | 39.6±0.8    | 39.8±0.5    |
> > >     |        | hopper              | 59.6±4.8   | 71.2±8.1  | 70.9±8.7    | 74.9±9.8    |
> > >     |        | walker2d            | 79.0±5.8   | 86.1±2.7  | 66.7±4.0    | 66.4±4.6    |
> > >     | **Average** |                | 60.4       | **66.8**      | 59.0        | 60.4       |
> > >     | **Average over all tasks** | | 47.0       | 56.0      | 53.3        | 55.3       |
> > >
> > >
> > > 2. **‘Have you considered evaluating diffusion-based methods in the corrupted data setting’**
> > >
> > >     Thank you for your insightful question. We agree that diffusion models hold significant potential for enhancing the robustness of offline RL. However, their application to addressing corrupted data remains relatively unexplored. For instance, while DMBP [1] utilizes diffusion models to manage observation perturbations during testing time, its performance would significantly degrade when trained on corrupted data. Typically, diffusion models in offline RL are trained using clean datasets, and adapting them to learn from corrupted datasets presents considerable challenges. The strong expressive capacity of diffusion models might cause them to fit a corrupted distribution, leading to undesired performance. Nonetheless, there have been some exciting developments in the computer vision domain [2][3] that provide promising insights and potential objective designs for diffusion models, which could guide valuable future research in this area.
> > >
> > >
> > > [1] DMBP: Diffusion model-based predictor for robust offline reinforcement learning against state observation perturbations[C]//The Twelfth International Conference on Learning Representations.
> > >
> > > [2] Daras G, Shah K, Dagan Y, et al. Ambient diffusion: Learning clean distributions from corrupted data[J]. Advances in Neural Information Processing Systems, 2024, 36.
> > >
> > > [3] Kawar B, Elata N, Michaeli T, et al. Gsure-based diffusion model training with corrupted data[J]. arXiv preprint arXiv:2305.13128, 2023.

---

### Official Review · Reviewer_4tSr · 2024-11-04

**Soundness:** 3
**Presentation:** 3
**Contribution:** 3
**Rating:** 6
**Confidence:** 3

**Summary:**

The work deals with the challenges posed by data corruption in offline RL and proposes a novel approach RDT. The primary contribution is the application of sequence modeling, specifically through a decision transformer, which has shown greater robustness against corrupted data compared to conventional RL methods that rely on temporal difference learning. The RDT model incorporates three key techniques—embedding dropout, Gaussian weighted learning, and iterative data correction—to enhance robustness. Empirical results indicate that RDT achieves superior performance under various corruption scenarios and provides resilience to both training-time data corruption and testing-time observation perturbations.

**Strengths:**

- Novel Method: The use of sequence modeling as a robust alternative to traditional temporal-difference-based methods for offline RL under corrupted data conditions is a significant and novel contribution.
- The paper introduces a robust architecture combining embedding dropout, Gaussian weighted learning, and iterative data correction, addressing different aspects of the corruption challenge.
- The experiments are extensive, covering several benchmarks and a variety of corruption scenarios, demonstrating consistent outperformance of RDT over baselines.
- The study highlights the potential of sequence modeling to handle real-world data corruption in offline RL, an area with increasing practical relevance.

**Weaknesses:**

-The datasets used are predominantly standard RL benchmarks. Incorporating more complex, real-world datasets could provide stronger support for the robustness claims in real-world applications.

- While the paper corrects actions and rewards through iterative data correction, it leaves out the correction of states, which might impact robustness in tasks with high state complexity.

- The study lacks a detailed comparison with other robust sequence modeling approaches outside of decision transformers, such as those incorporating value functions or other regularization techniques.

**Questions:**

Could the authors provide more clarity on the choice of dropout probabilities for different embeddings (states, actions, rewards) and its sensitivity to these values? How were these values chosen in the experiments?

The paper states that actions and rewards are corrected iteratively but leaves out state correction. What challenges or limitations prevented state correction? Would the authors consider incorporating state correction as future work, and how would it theoretically affect robustness?

---

> ### Author Response · Authors · 2024-11-20
> **Author Response**
>
> Thank you for the valuable comments. We provide clarification on your concerns as follows. We hope these new experimental results and explanations can address your concerns, and we look forward to your further feedback.
>
> 1. **‘Incorporating more complex, real-world datasets could provide stronger support for the robustness claims in real-world applications’. (W1)**
>
>     Thank you for the valuable suggestion. Our primary goal is to demonstrate the effectiveness of robust sequence modeling method on popular offline RL benchmarks [1], including **MuJoCo, Kitchen, and Adroit. Specifically, the Kitchen and Adroit tasks are two challenging and relatively realistic robotic manipulation tasks.**
>
>     To further demonstrate the robustness of RDT in real-world applications, we evaluate it on a near real-world benchmark, the NeoRL benchmark [2], with the *citylearn-medium* and *finance-medium* datasets. The Finance dataset simulates real stock market trading, while CityLearn focuses on optimizing energy storage across various buildings to reshape electricity demand.  We randomly sample 20% and 10% of the Finance and CityLearn datasets, respectively, resulting in both datasets having a similar size of approximately $2 \times 10^4$ transitions.
>
>     We compare our method against baselines such as RIQL, DeFog, and DT. We have included these experiments in Appendix D.12 and in the table below for your reference. Notably, **RDT achieves the best average performance compared to the other baselines, outperforming DT by 12.13% under Finance dataset.** This result further demonstrates its robustness and promise for real-world applications.
>
>     | Task                   |    Attack    |    DeFog    |    RIQL     |     DT     |   **RDT**   |
>     |------------------------|--------------|-------------|-------------|------------|-------------|
>     | citylearn (20\%)  | State  | **39138.9**  |  30235.3  | 37878.9  | 39124.2    |
>     |                          | Action | 38970.6   |  29093.2  | 35589.0  | **39066.4**    |
>     |                          | Reward | 38156.4  |  29634.8  | 37626.9  |  **38909.8**  |
>     | Average          | |  38755.3  | 29654.4  | 37031.6  | **39033.5**   |
>     | finance (10\%)      | State  |  92.4   |    89.4   |    85.7   | **104.5**   |
>     |                          | Action |  96.4   |   **100.6**  |    91.4   | 99.8    |
>     |                          | Reward |  90.2  |    89.2   |    89.7   |  **95.1**   |
>     | Average      |  | 93.0   | 93.1   | 89.0   | **99.8**   |
>
>
> 2. **‘What challenges or limitations prevented state correction’. (W2,Q2)**
>
>     We have included a discussion on the state prediction and correction mechanisms in Appendix D.14. We compare two variants: DT(SP), which predicts states based on the original DT, and DT(SP) w. IDC, which applies iterative state correction to DT(SP). These variants are evaluated under state corruption.
>
>     Figure 19 shows that DT(SP) does not consistently offer benefits over the original DT, and it sometimes even results in a performance drop. Additionally, DT(SP) w. IDC fails to achieve satisfactory performance across different $\zeta$ values. To provide a convenient reference, we format the following table extracted from Figure 19.
>
>
>     | Task | DT| DT(SP) | DP(SP) w. IDC ($\zeta$=1.0) | DP(SP) w. IDC ($\zeta$=3.0) | DP(SP) w. IDC ($\zeta$=5.0) |
>     | -----| ---|---|---|---|----|
>     | MuJoCo (10%) | 20.1 | 23.2 | 23.1 | 24.5 | 22.8 |
>     | KitChen      | 33.2 | 25.2 | 14.5 | 23.4 | 24.0 |
>     | Adorit (1%)  | 84.6 | 81.9 | 76.3 | 77.2 | 79.1 |
>
>
>     Several factors may contribute to the challenges of state prediction and correction:
>
>     1. **High Dimensionality of States**: The high dimensionality of states makes it difficult to predict states accurately. For example, in KitChen tasks, the state space dimension (60) is much higher compared to the action space dimension (9) and reward dimension (1).
>     2. **Data Corruption**: In our setting, data corruption can exacerbate the challenge of predicting states, resulting in further declines in performance.
>
>     Therefore, we leave the state prediction and correction as promising future work. Advanced state representation techniques, such as discrete representation learning through VQVAE [3] and contrastive learning [4], could potentially be beneficial in addressing this issue.

---

> ### Author Response · Authors · 2024-11-20
> **Author Response**
>
> 3. **‘a detailed comparison with other robust sequence modeling approaches outside of decision transformers’. (W3)**
>
>     We have included several offline RL algorithms in our main results that incorporate regularization or robustness techniques. Specifically, **UWMSG [5]** and **RIQL [6]** are methods that enhance value functions with additional robust mechanisms like uncertainty weighting and ensemble learning. In addition, **DeFog[7]** is a robust sequence modeling algorithm robust to frame-dropping in offline RL.
>
>     Moreover, we also combine robust offline RL (e.g., UWMSG and RIQL) methods with transformer policy backbone in Section 4.5. These two variants simultaneously incorporate value functions and sequence modeling, serving as a strong baseline. Our RDT outperforms all these methods under various data corruption settings, demonstrating its superior ability to handle data corruption effectively.
>
> 4. **‘more clarity on the choice of dropout probabilities** **$p$’. (Q1)**
>
>     As stated on line 248 at the end of Section 3.2.1, we use the same dropout probabilities $p$ for different element (state, action, and reward) embeddings. We also conduct an ablation study on these probabilities in Appendix D.3. Setting dropout probabilities $p$ within a moderate range (0.1 to 0.3) can enhance performance. However, higher dropout probabilities can degrade performance, likely due to excessive loss of information from the input.
>
>
> [1] D4rl: Datasets for deep data-driven reinforcement learning[J]. arXiv preprint arXiv:2004.07219, 2020.
>
> [2] NeoRL: A near real-world benchmark for offline reinforcement learning[J]. Advances in Neural Information Processing Systems, 2022, 35: 24753-24765.
>
> [3] Neural discrete representation learning[J]. Advances in neural information processing systems, 2017, 30.
>
> [4] Contrastive learning as goal-conditioned reinforcement learning[J]. Advances in Neural Information Processing Systems, 2022, 35: 35603-35620.
>
> [5] Corruption-robust offline reinforcement learning with general function approximation[J]. Advances in Neural Information Processing Systems, 2024, 36.
>
> [6] Towards robust offline reinforcement learning under diverse data corruption[J]. arXiv preprint arXiv:2310.12955, 2023.
>
> [7] Decision transformer under random frame dropping[J]. arXiv preprint arXiv:2303.03391, 2023.

---

> ### Author Response · Authors · 2024-11-25
>
> We sincerely thank you for your valuable suggestions and time spent on our work. We hope that we have addressed your main questions in the rebuttal. As the due date (November 26 at 11:59pm AoE) for the discussion period is approaching, we hope to receive your further feedback and address your concerns once again. We are committed to addressing any concerns or suggestions you may have to enhance the manuscript.

---

> > ### Comment · Reviewer_4tSr · 2024-11-28
> > **Response**
> >
> > Thanks for the authors responding some of the issues. I'll keep my score unchanged.

---

### Author Response · Authors · 2024-11-20
**Common Response**

We sincerely thank the reviewers for their valuable comments. We have revised our paper to address the concerns raised, with modifications highlighted in blue for ease of reading. Specifically, we have made the following key improvements to our paper:

1. Provided detailed results with uncertainty measures for the main experiments on random and adversarial attacks in Appendix D.6.
2. Included new results of RDT under various dataset scales in Appendix D.10.
3. Added results of RDT under different data quality levels (medium and expert) for MuJoCo in Appendix D.11.
4. Presented new results of RDT on the near real-world dataset, NeoRL, in Appendix D.12.
5. Added new ablation studies on the three components, featuring variants of RDT w/o (ED, GWL, IDC), in Appendix D.9.
6. Conducted additional evaluation of Iterative Data Correction in Appendix D.13.
7. Discussed state prediction and correction in Appendix D.14.
8. Provided a discussion of RDT's limitations in Appendix E.
9. Revised the manuscript to include more explanations of important terms and hyperparameters.

We hope these new experimental results and modifications address the reviewers' concerns. If you have any additional questions or concerns, please feel free to post them, and we will be glad to discuss them.

---

### Meta-Review · Area_Chair_rcj3 · 2024-12-19

**Metareview:**

This paper proposes to use sequence models to tackle data corruption in offline RL with a limited dataset. It introduces Robust Decision Transformer with three componenets embedding dropout, Gaussian weighted learning, and iterative data correction. Extensive experiments on diverse datasets show the robustness and superior results of the proposed method against TD-based approaches.

Strengths:
- Clear presentation
- Novel use of decision transformers as a robust modeling method in offline RL
- Comprehensive empirical evaluation across various datasets, data scale and corruption scenarios.

Weaknesses:
- Limited scope to offline RL with a limited dataset and data corruption. The performance is comparable with TD-based methods at the full dataset scale.
- Challenges in applying state correction. This is left as a future direction.

Based on the novelty and robust improvement of the RDT method in a diverse setting, the reviewers reach unanimous agreement of accepting this submission.

**Additional Comments On Reviewer Discussion:**

There were various concerns / questions around evaluations in the initial reviews including:
- Lack of comparison with baselines
- Request of additional experiment setting
- Lack of ablation
The authors provided additional experiments during the rebuttal period. Most of the concerns have been addressed.

I would encourage the authors to incorporate their additional explanation and experiments in the final revision, and clarify its limitation.

---

### Decision · Program_Chairs · 2025-01-22

Accept (Poster)